# Bridging the spatial gaps of the Ammonia Monitoring Network using satellite ammonia measurements

**Rui Wang[1], Da Pan[1], Xuehui Guo[1], Kang Sun[2,3], Lieven Clarisse[4], Martin Van Damme[4,5], Pierre-François Coheur[4], Cathy Clerbaux[4,5,6], Melissa Puchalski[7], and Mark A. Zondlo[1]**

[1]Department of Civil and Environmental Engineering, Princeton University, Princeton, NJ, USA
[2]Department of Civil, Structural and Environmental Engineering, University at Buffalo, Buffalo, NY, USA
[3]Research and Education in eNergy, Environment and Water (RENEW) Institute,
University at Buffalo, Buffalo, NY, USA
[4]Spectroscopy, Quantum Chemistry and Atmospheric Remote Sensing (SQUARES),
Université libre de Bruxelles (ULB), Brussels, Belgium
[5]Royal Belgian Institute for Space Aeronomy, Brussels, Belgium
[6]LATMOS/IPSL, Sorbonne Université, UVSQ, CNRS, Paris, France
[7]Office of Air and Radiation, US Environmental Protection Agency, Washington, DC, USA

**Correspondence:** Mark A. Zondlo (mzondlo@princeton.edu)

**Abstract.** Ammonia ($NH_3$) is a key precursor to fine particulate matter ($PM_{2.5}$) and a primary form of reactive nitrogen. The limited number of $NH_3$ observations hinders the further understanding of its impacts on air quality, climate, and biodiversity. Currently, $NH_3$ ground monitoring networks are few and sparse across most of the globe, and even in the most established networks, large spatial gaps exist between sites and only a few sites have records that span longer than a decade. Satellite $NH_3$ observations can be used to discern trends and fill spatial gaps in networks, but many factors influence the syntheses of the vastly different spatiotemporal scales between surface network and satellite measurements. To this end, we intercompared surface $NH_3$ data from the Ammonia Monitoring Network (AMoN) and satellite $NH_3$ total columns from the Infrared Atmospheric Sounding Interferometer (IASI) in the contiguous United States (CONUS) and then performed trend analyses using both datasets. We explored the sensitivity of correlations between the two datasets to factors such as satellite data availability and distribution over the surface measurement period, as well as agreement within selected spatial and temporal windows. Given the short lifetime of atmospheric ammonia and consequently sharp gradients, smaller spatial windows show better agreement than larger ones except in areas of relatively uniform, low concentrations where large windows and more satellite measurements improve the signal-to-noise ratio. A critical factor in the comparison is having satellite measurements across most of the measurement period of the monitoring site. When IASI data are available for at least 80 % of the days of AMoN's 2-week sampling period within a 25 km spatial window of a given site, IASI $NH_3$ column concentrations and the AMoN $NH_3$ surface concentrations have a correlation of 0.74, demonstrating the feasibility of using satellite $NH_3$ columns to bridge the spatial gaps existing in the surface network $NH_3$ concentrations. Both IASI and AMoN show increasing $NH_3$ concentrations across the CONUS (median: 6.8 % yr$^{-1}$ versus 6.7 % yr$^{-1}$) in the last decade (2008–2018), suggesting the $NH_3$ will become a greater contributor to nitrogen deposition. $NH_3$ trends at AMoN sites are correlated with IASI $NH_3$ trends ($r = 0.66$) and show similar spatial patterns, with the highest increases in the Midwest and eastern US. In spring and summer, increases in $NH_3$ were larger than 10 % yr$^{-1}$ in the eastern US and Midwest (cropland dominated) and the western US (pastureland dominated), respectively. $NH_3$ hotspots are defined as regions where the IASI $NH_3$ column is larger than the 95th percentile of the 11-year CONUS map ($6.7 \times 10^{15}$ molec. cm$^{-2}$), they also experience increasing concentrations over time, with a median of $NH_3$ trend

of $4.7\,\%\,\mathrm{yr}^{-1}$. IASI data show large $NH_3$ increases in urban areas ($8.1\,\%\,\mathrm{yr}^{-1}$), including 8 of the top 10 most populous regions in the CONUS, where AMoN sites are sparse. A comparison between IASI $NH_3$ concentration trends and state-level $NH_3$ emission trends is then performed to reveal that positive correlations exist in states with strong agricultural $NH_3$ emissions, while there are negative correlations in states with low $NH_3$ emissions and large $NO_x$ emissions, suggesting the different roles of emission and partitioning in $NH_3$ increases. The increases in $NH_3$ could have detrimental effects on nearby eco-sensitive regions through nitrogen deposition and on aerosol chemistry in the densely populated urban areas, and therefore they should be carefully monitored and studied.

## 1   Introduction

Gas-phase ammonia ($NH_3$) is the most abundant alkaline gas in the atmosphere, mainly emitted from agricultural activities such as nitrogen fertilizer applications and livestock waste volatilization (Bouwman et al., 1997; Paulot et al., 2014). As a major precursor to fine particulate matter ($PM_{2.5}$), $NH_3$ critically affects aerosol heterogeneous chemistry, air quality, visibility, human health, and climate (Hauglustaine et al., 2014; Hill et al., 2019; Lawal et al., 2018; Malm et al., 2004). Ammonia neutralizes sulfuric acid ($H_2SO_4$) and nitric acid ($HNO_3$) in the atmosphere to form ammoniated aerosols, ammonium sulfate (($NH_4$)$_2SO_4$), and ammonium nitrate ($NH_4NO_3$), which in total can contribute to more than $50\,\%$ of total $PM_{2.5}$ mass (Feng et al., 2020). $NH_4NO_3$ is critical during wintertime haze periods because the cold and humid condition favor its formation (Shah et al., 2018; Zhai et al., 2021). Moreover, $NH_3$ plays an important role in the nitrogen cycle. Wet deposition of $NH_4^+$ dominates the wet inorganic nitrogen deposition at nearly $70\,\%$ of monitoring sites in the United States (Li et al., 2016). Total $NH_x$ ($\equiv NH_3\,(\mathrm{g}) + NH_4^+\,(\mathrm{aq})$) deposition is expected to become even more dominant in the future because $NO_x$ emissions are decreasing under new pollution controls, while $NH_3$ emissions are predicted to continue to increase with the rising global food demands (Erisman et al., 2008; Goldberg et al., 2021; Pinder et al., 2008). Excessive $NH_3$ deposition in the non-agricultural ecosystems can reduce biodiversity, result in soil acidification, and increase eutrophication, especially in sensitive ecosystems (Ellis et al., 2013; Phoenix et al., 2006). Although $NH_3$'s importance has been well recognized, routine $NH_3$ observations are lacking even in countries with comprehensive monitoring networks, partly due to the difficulty of measuring gas-phase $NH_3$ (von Bobrutzki et al., 2010; Fehsenfeld et al., 2002). The Ammonia Monitoring Network (AMoN) (Puchalski et al., 2015) is the only routine set of $NH_3$ measurements in the United States, with 110 active AMoN sites in the contiguous United States (CONUS) in 2021, providing high-quality surface observations of $NH_3$. AMoN data have been used widely for model evaluation and long-term trend analysis (Butler et al., 2016; Nair et al., 2019; Yao and Zhang, 2016, 2019). AMoN only provides biweekly $NH_3$ observations, in contrast to monitoring networks for two other important gas-phase precursors of $PM_{2.5}$, $SO_2$ and $NO_2$, which provide hourly- or daily-scale observations. $PM_{2.5}$, $SO_2$, and $NO_2$ are directly regulated as criteria pollutants; however, contributions from $NH_3$ emissions sources must be considered in State Implementation Plan (SIP) demonstrations for areas out of attainment for $PM_{2.5}$, which can be a challenge for areas lacking $NH_3$ measurements (EPA Air Quality Implementation Plans, 2023).

Population-weighted $PM_{2.5}$ concentrations are widely used to estimate the health effects of $PM_{2.5}$; however, the sparse number of $NH_3$ sites with only biweekly or monthly resolution makes it difficult to derive population-weighted $PM_{2.5}$ precursor datasets. Gas-phase $NH_3$ is critical to determine the partitioning of the total $NH_x$ (Hennigan et al., 2015), and the lack of gas-phase $NH_3$ observations hampers the evaluation of chemistry models. The ISORROPIA II thermodynamic model has been extensively adopted to compute the equilibrium composition for the inorganic aerosol systems (Fountoukis and Nenes, 2007) and requires both gas- and aerosol-phase data as input to provide accurate and robust results (Hennigan et al., 2015). However, the limited number of $NH_3$ ground monitoring sites currently prevents synthesizing the AMoN $NH_3$ data with other ground monitoring networks, e.g., the Interagency Monitoring of Protected Visual Environments (IMPROVE), as input for ISORROPIA II (Pan et al., 2020). GEOS-Chem implemented with ISORROPIA II was found to significantly underestimate gas-phase $NH_3$ and overestimate $NH_4^+$ in winter (Holt et al., 2015; Nair et al., 2019; Walker et al., 2012), with the normalized $NH_4^+$ mean biases as high as $86\,\%$ in January at sites for IMPROVE (Holt et al., 2015). The lifetime of $NH_3$ ranges from hours to days; hence large spatiotemporal variability exists (Golston et al., 2020; Miller et al., 2015; Wang et al.; 2021), and large spatial gaps exist in the current AMoN. Currently there are no AMoN sites in some states, e.g., North Dakota and South Dakota, and only 12 sites are within the characteristic length scale ($12\,\mathrm{km}$) of $NH_3$ hotspot regions (Wang et al., 2021). A total of 10 national parks in the US are within $100\,\mathrm{km}$ of an $NH_3$ hotspot, and more observations are needed to quantify the impacts of these hotspots on dry $NH_3$ deposition in these regions (Pan et al., 2021). A lack of long-term AMoN data also hinders the possibility of investigating $NH_3$ trends in the CONUS. Increasing $NH_3$ concentrations

are observed using AMoN data, yet all of the previous trend analyses are limited to fewer than 20 AMoN sites that may not be representative of $NH_3$ trends in the CONUS (Butler et al., 2016; Yao and Zhang, 2016, 2019).

Satellite $NH_3$ observations are on a global and daily basis, providing long-term trends and ubiquitous coverage. Instruments that measure $NH_3$ include the Infrared Atmospheric Sounding Interferometer (IASI) on the MetOp satellites, the Cross-track Infrared Sounder (CrIS) on the NOAA and NASA Suomi National Polar-orbiting Partnership (S-NPP) and on the Joint Polar Satellite System-1 and System-2 (JPSS-1 and JPSS-2) satellites, the Tropospheric Emission Spectrometer (TES) on the NASA Aura satellite, the Atmospheric Infrared Sounder (AIRS) on the NASA EOS Aqua satellite, and the Thermal and Near Infrared Sensor for Carbon Observations – Fourier Transform Spectrometer (TANSO-FTS) on the Greenhouse Gases Observing SATellite (GOSAT) (Clarisse et al., 2009; Shephard et al., 2011; Shephard and Cady-Pereira, 2015; Someya et al., 2020; Warner et al., 2016). Satellite $NH_3$ data have been widely used to constrain $NH_3$ emissions, estimate $NH_3$ deposition, and analyze $NH_3$ trends (H. Cao et al., 2020, 2022; Chen et al., 2021 TS1; Kharol et al., 2018; Van Damme et al., 2021). Van Damme et al. (2021) utilized 11-year IASI $NH_3$ observations and found a worldwide $NH_3$ increase ($12.8 \pm 1.3\%$) from 2008 to 2018 with especially large increases in east Asia ($75.7 \pm 6.3\%$) and North America ($26.8 \pm 4.5\%$). Warner et al. (2017) used 14-year AIRS $NH_3$ measurements and found a statistically significant $NH_3$ increase ($2.61\%\,yr^{-1}$) in the US from 2002 to 2016.

The global daily coverage and long-term data record make it possible for satellite observations to fill the spatial and temporal gaps of the current ground monitoring networks. Although limited in numbers, the validations of satellite $NH_3$ observations with in situ measurements provide confidence in integrating the two datasets (Guo et al., 2021; Sun et al., 2015). Sun et al. (2015) performed the first daily- and pixel-scale satellite $NH_3$ validations using TES $NH_3$ columns and airborne $NH_3$ observations in the San Joaquin Valley of California, USA, showing that the differences between the total $NH_3$ column and the in situ total column were within 6 %. However, the validation included only 9 TES pixels, and TES is no longer in operation. Guo et al. (2021) showed that IASI $NH_3$ columns and $NH_3$ columns derived from airborne and ground-based $NH_3$ observations were indistinguishable from one another on a daily and pixel basis in Colorado, USA, in summer. All of these validation works were carried out in specific seasons and were limited to source regions with high $NH_3$ concentrations (Guo et al., 2021; Sun et al., 2015; Warner et al., 2016). Ground-based FTIR $NH_3$ observations provided a better temporal coverage for evaluating IASI and CrIS $NH_3$ retrievals; however, low-concentration sites were excluded from the evaluation, and only $\sim 10$ sites were included across the globe (Dammers et al., 2016; Dammers et al., 2017). Furthermore, FTIR-based measurements also have not been directly validated against in situ measurements of $NH_3$ vertical profiles.

To capitalize on the benefits of both surface and satellite observations and synthesize these datasets, a detailed understanding of the comparison between IASI $NH_3$ column concentrations and AMoN $NH_3$ surface concentrations is necessary. Here we focus on IASI $NH_3$ measurements because it offers the longest data record (2008–present) among the satellite $NH_3$-measuring instruments. The comparison between AMoN and IASI is complex because AMoN is a ground-based point measurement integrated over 14 d, whereas IASI is a space-borne volumetric measurement averaged over the pixel footprint at the instantaneous overpass time. There are several factors that need to be taken into consideration.

1. *The extent to which the IASI $NH_3$ column represents the surface AMoN $NH_3$ concentration.* Knowledge of $NH_3$ vertical profiles in the atmosphere is limited due to the lack of observational data, and model-simulated $NH_3$ vertical profiles are often biased compared with the airborne measurements (Schiferl et al., 2016). Ammonia is mostly concentrated in the planetary boundary layer (PBL) because of its short lifetime ($\sim$ hours to days) and surface emission sources (Dentener and Crutzen, 1994; Guo et al., 2021; Sun et al., 2015; Seinfeld and Pandis, 2016). Sun et al. (2015) showed that $NH_3$ was almost well mixed in the lower PBL, and the TES $NH_3$ columns were strongly correlated ($R^2 = 0.82$) with the median $NH_3$ mixing ratios measured at the surface, demonstrating that satellite $NH_3$ columns could represent the ground $NH_3$ concentrations. Van Damme et al. (2015) converted IASI $NH_3$ columns to surface $NH_3$ concentrations using fixed $NH_3$ profiles generated by GEOS-Chem and then performed monthly comparisons with ground monitoring networks. IASI-derived surface $NH_3$ observations are in fair agreement with ground observations in Europe, China, and Africa but are limited to a small number of sites in each region for a short time range, e.g., 27 sites in Europe in 2011 (Van Damme et al., 2015). Furthermore, the latest IASI $NH_3$ products have switched to a new algorithm and no longer use a fixed $NH_3$ profile (Whitburn et al., 2016; Van Damme et al., 2017).

2. *Optimal spatial window for comparing and integrating satellite pixels and AMoN sites.* Previous comparisons of satellite $NH_3$ retrievals with observations from ground monitoring networks simply averaged the data from the monitoring site within a coarse model grid ($\sim 100\,km$) with the averaged modeling/satellite $NH_3$ concentration of the whole grid (Kharol et al., 2018; Nair et al., 2019; Van Damme et al., 2015). If $NH_3$ concentrations are uniformly distributed within the spatial window, increasing the spatial window will

increase the number of IASI pixels and decrease the signal-to-noise ratio. However, the spatial heterogeneity of $NH_3$ is quite large near hotspots due to its short lifetime (Golston et al., 2020; Miller et al., 2015; Wang et al., 2021; Warner et al., 2016). The relationship between spatial window size and satellite and surface measurement agreement needs to be examined in more detail.

3. *Temporal distribution of satellite measurements across the 2-week AMoN sampling period.* Previous comparisons of model or satellite products against surface observations did not consider the distribution of IASI measurements during the 2-week sampling period (Kharol et al., 2018; Nair et al., 2019; Van Damme et al., 2015). AMoN measures continuously, whereas a series of cloudy days would preclude any valid satellite measurements. Therefore, any AMoN and satellite comparison is intrinsically biased towards clear-sky days on the satellite side but includes all conditions for the AMoN site.

4. *Number of available IASI pixels in the comparison.* Guo et al. (2021) have shown that, even at low column amounts, IASI $NH_3$ has no known biases. AMoN is an extremely sensitive measurement of $NH_3$, far more precise than any satellite $NH_3$ product (NADP, 2023; Van Damme et al., 2017). Therefore, increasing the number of satellite measurements within a certain spatiotemporal window is expected to improve the signal-to-noise ratio in the satellite measurements and may lead to improved agreements with AMoN under clean conditions.

5. *Regional and seasonal variabilities.* Different regional and seasonal patterns are expected to influence the comparison. The performances of thermal infrared sounders are highly affected by the thermal contrast between the surface air temperature and skin temperature (Clarisse et al., 2010). In winter, low thermal contrast results in low sensitivity, which explains the low number of IASI pixels in winter compared to summer (Clarisse et al., 2010; Guo et al., 2021). Kharol et al. (2018) showed that CrIS surface $NH_3$ concentrations had an overall mean CrIS–AMoN difference of $\sim +15\%$; however, they only averaged CrIS data over the warm season in 2013.

In this study, to demonstrate the capabilities of using IASI $NH_3$ observations to augment the ground monitoring network, we performed a comprehensive comparison between IASI and AMoN on biweekly and seasonal scales. We directly compare the correlation between IASI $NH_3$ columns with AMoN surface $NH_3$. We avoided converting column $NH_3$ into surface concentrations because of possible biases introduced by assuming vertical profiles, boundary layer heights at local sites, and gas-phase and aerosol partitioning. The impacts of the different factors on the comparison are examined in the context of the points raised above. After identifying the most optimal method for comparison, we examined $NH_3$ trends over AMoN sites and the larger applicability of using satellite retrievals to discern $NH_3$ trends over regions and seasons lacking AMoN data.

## 2    Data and methods

### 2.1    Satellite $NH_3$ observations

IASI is an infrared sounder deployed on board the MetOp-A, MetOp-B, and MetOp-C platforms in sun-synchronous orbits since October 2006, September 2012, and November 2018, respectively. IASI has a swath of 2200 km and provides global coverage twice per day at around 09:30 and 21:30 mean local solar time. At nadir, the IASI footprint has a 12 km diameter. The first IASI $NH_3$ product was developed by Clarisse et al. (2009) by converting the brightness temperature differences into total $NH_3$ columns. Later on, a flexible and robust retrieval algorithm based on an artificial neural network for IASI (ANNI) (Whitburn et al., 2016) was developed. The latest version is a reanalyzed dataset that uses the European Centre for Medium-Range Weather Forecasts Reanalysis v5 (ERA5) as its meteorological input (Van Damme et al., 2017; Van Damme et al., 2021). Because these meteorological data are coherent in time, the reanalysis dataset is the most appropriate dataset to study trends. For the present analyses, we used IASI version 3.1 reanalysis (v3.1r) retrieval product data from the MetOp-A (2008–2018) and MetOp-B (2013–2018) satellites (limited to cloud fraction $\leq 25\%$). Only the morning orbits were analyzed because of higher sensitivity than the evening overpasses (Clarisse et al., 2010).

### 2.2    Ground-based observations

AMoN is the only network providing a consistent, long-term record of $NH_3$ gas concentrations across the United States. AMoN was established by the National Atmospheric Deposition Program (NADP) in October 2007 and expanded to 19 sites in 2010 and 105 sites in 2018. AMoN deploys Radiello® passive samplers that rely upon diffusion theory, where gas-phase $NH_3$ is adsorbed onto a cylindrical interior filter and extracted as $NH_4^+$ to be analyzed by flow injection analysis (FIA). AMoN provides biweekly surface $NH_3$ concentrations, and the network detection limit is $0.083\,mg\,NH_4^+\,L^{-1}$ ($\sim 0.078\,\mu g\,NH_3\,m^{-3}$) for the 2-week samples in 2020 (NADP, 2023). The Radiello® passive samplers were found to be biased low by 37 % against denuders used as the reference method (Puchalski et al., 2011). In this study, we are comparing the relative variations instead of absolute concentrations of IASI and AMoN; therefore, the low bias of AMoN measurements is not as relevant to the outcome.

We incorporated data from all AMoN sites with one notable exception. Using satellite imagery, we determined that the AMoN site in Logan, Utah (UT01), is located only $\sim 100$ m away from a livestock farm. Ammonia concentrations downwind of a beef/dairy feedlot at this distance are far above background levels and unrepresentative of those at the local–regional scales (1–10 km) (Golston et al., 2020; Miller et al., 2015; Sun et al., 2018). Concentrations at UT01 are expected to be strongly dependent upon the extent to which local winds blow directly from that farm to the AMoN site throughout the 2-week integration period. Not surprisingly, the UT01 site has the highest annual mean concentration (16.2 µg m$^{-3}$) in the entire AMoN network (3 times higher than the next one). Furthermore, this AMoN site may be particularly susceptible to trends in animal operations or management practices at the farm. While it is possible the measurements of UT01 are representative of the local region, it is beyond the scope of this work to make such an assessment of its representativeness. For trend analyses, we only include AMoN sites with full-year coverage during 2008–2018 ($N = 13$).

## 2.3 Trend analyses

### 2.3.1 Oversampled NH$_3$ maps for trend analysis

From 2008 to 2018, a $0.02° \times 0.02°$ ($\sim 2$ km) annual mean NH$_3$ map of the CONUS was created each year based on a physical oversampling algorithm that represents the satellite spatial response functions as generalized 2-D super-Gaussian functions (Sun et al., 2018). This algorithm weighs IASI measurements by their uncertainties, which include varying sensitivities to thermal contrast as described in Sun et al. (2018) and Wang et al. (2021). To evaluate the seasonal trends, for each year, seasonally averaged oversampling maps were also generated for spring (March, April, and May, MAM), summer (June, July, and August, JJA), autumn (September, October, and November, SON), and winter (December, January, and February, DJF). For each season, we were able to achieve sufficiently overlapped IASI pixels through calculating the sum of the unnormalized spatial response function (SRF) of the oversampling results. A large sum of unnormalized SRF means the level 3 grid is covered by more level 2 pixels. Sun et al. (2018) and Wang et al. (2021) have a detailed description of SRF. The oversampling products are only used for the trend analyses in Sect. 4 to achieve a high spatial resolution. For IASI and AMoN comparison results in Sect. 3, the oversampling products are not used, since it sacrifices the temporal resolution.

### 2.3.2 Mann–Kendall test and Theil–Sen's slope estimator for trend analysis

We use the Mann–Kendall (MK) test and Theil–Sen's slope estimator for NH$_3$ trend analyses. The non-parametric Mann–Kendall test and Theil–Sen's slope estimator are widely used in detecting trends of variables in meteorological and hydrological fields (Ahn and Merwade, 2014; Kendall, 1975; Yue and Wang, 2004). The Kendall rank correlation coefficient, commonly referred to as Kendall's $\tau$ coefficient, is a statistic used to measure the rank correlation. An MK test is a non-parametric hypothesis test for statistical dependence based on Kendall's $\tau$ coefficient. The Theil–Sen's slope estimator is commonly used to fit a line to data points by calculating the median of the slopes of all lines through pairs of points.

Unlike simple linear regression, the Mann–Kendall test and Theil–Sen's slope estimator do not require the data to follow normal distribution and therefore are more robust to any outliers (Yue and Wang, 2004). This method is computationally efficient and is insensitive to outliers. For skewed and heteroscedastic data, the Theil–Sen estimator can be significantly more accurate than linear least squares regression. For normally distributed data, the Theil–Sen estimator competes well against the least squares in terms of statistical power (Yue and Wang, 2004).

In this study, Theil–Sen's slope was used to estimate 2008–2018 NH$_3$ trends, and the MK test was used to derive the significance level of trends.

## 3 IASI and AMoN comparison

### 3.1 Sensitivity to spatial windows

For the initial analysis, we first used the simplest method for comparing the satellite measurements with ground observations. In other words, for each AMoN site, we average all IASI observations within a given radius of the AMoN site during the sampling time frame (2 weeks) for comparison and refer to that radius as a spatial window. We define each AMoN sample with co-located IASI pixels as an AMoN–IASI pair. If the distribution of NH$_3$ pixels is spatially uniform, increasing the spatial window may improve the correlation between the two datasets because of a larger number of IASI pixels. Larger spatial windows include more IASI pixels than smaller spatial windows but at the expense of potentially not being representative of the AMoN site. In addition, a larger region is likely to encompass NH$_3$ spatial gradients. In contrast, small spatial windows may only include a limited number of IASI pixels, encompassing more inherent noise in the satellite measurements, especially if close to the detection limit. Each integrated 2-week AMoN measurement for each site was correlated with any relevant satellite data within the spatial window (total of 104 AMoN sites with 16 093 measurements). Correlations between IASI and AMoN for different spatial windows (15, 25, 50, and 100 km) are summarized in Table 1. The minimum spatial window radius of 15 km is based upon an approximate scale for NH$_3$ hotspots (Wang et al., 2021).

As the spatial window becomes larger, mean temporal coverage (defined as the percentage of days with available IASI

**Table 1.** AMoN and IASI comparison results for different spatial windows.

| Spatial window | 15 km | 25 km | 50 km | 100 km |
|---|---|---|---|---|
| Pearson's $r$ | 0.35 | 0.41 | 0.45 | 0.44 |
| Mean temporal coverage per pair (%) | 31 | 44 | 57 | 71 |
| Mean no. IASI pixels per AMoN–IASI pair | 7 | 17 | 69 | 278 |
| No. AMoN–IASI pairs | 14 734 | 15 543 | 15 933 | 16 022 |

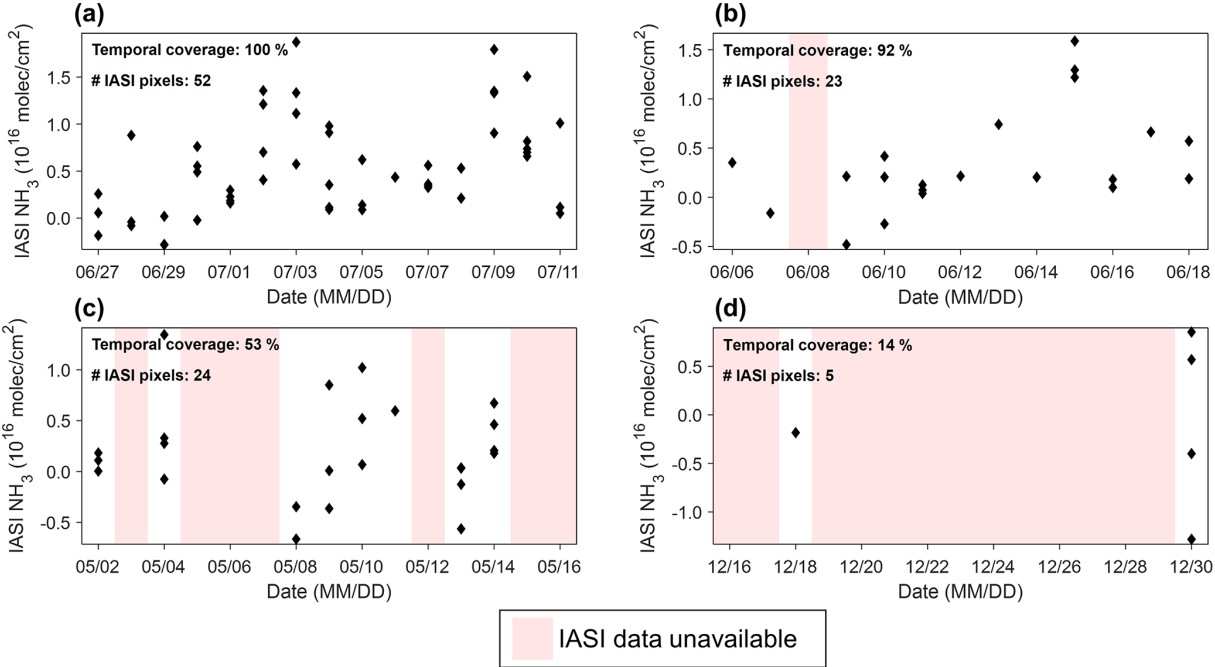

**Figure 1.** Examples of IASI data temporal coverage over the biweekly AMoN sampling period for an AMoN site in Yosemite National Park, California (CA 44): **(a)** several IASI measurements every day during the 2-week sampling period, **(b)** a few IASI measurements for most days of the 2-week sampling period, **(c)** many IASI measurements but only on several days during the 2-week sampling period, and **(d)** sparse IASI measurements for only several days during the 2-week sampling period.

data of the 2-week AMoN sampling period) and the number of IASI pixels both have significant increases, but Pearson's $r$ coefficient only increases slightly from 0.35 at a 15 km spatial window to 0.44 at a 100 km spatial window. Indeed, doubling the spatial window from 50 to 100 km yields an almost tripled mean number of IASI pixels, yet it maintains almost the same correlation with $r = 0.45$ and $r = 0.44$, respectively. This indicates that including IASI pixels at longer distances from the AMoN site may not be representative of the AMoN site, especially near sources or in regions with complex topography. The slightly increased $r$ value over spatial window range may result from a tradeoff between averaging spatial gradients versus integrating a larger number of IASI pixels to improve the signal-to-noise ratio of the satellite measurements. To balance these competing effects, we select 25 km as the nominal spatial window for the further comparisons.

## 3.2  Sensitivities to temporal coverage and the number of IASI pixels

$NH_3$ is a short-lived species with a complicated diurnal profile (Nair and Yu, 2020) and the potential for large day-to-day concentration changes because of the variability in emissions, wind speed, temperature, PBL height, and aerosol partitioning (Golston et al., 2020; Miller et al., 2015). Thus, the temporal distribution of satellite measurements within the AMoN measurement period may impact the comparison. Figure 1 illustrates four examples where the number of IASI pixels and their relative distribution throughout the 2-week AMoN integration period (using a 25 km spatial window) could affect the results. An ideal comparison case would have a uniform number of IASI measurements on each day during the approximate 14 d AMoN measurement period, similar to the case shown in Fig. 1a. In this case, there is no specific day having more weight than the other when calculating the biweekly mean. More common, however, are

**Table 2.** The impact of IASI data's temporal coverage for the 2-week AMoN sampling period (25 km spatial window).

| IASI temporal coverage per AMoN–IASI pair (%) | [0, 20) | [20, 50) | [50, 80) | [80, ∞) |
|---|---|---|---|---|
| $r$ | 0.17 | 0.29 | 0.47 | 0.74 |
| Mean no. IASI pixels per AMoN–IASI pair | 3 | 13 | 26 | 38 |
| No. AMoN–IASI pairs | 1766 | 7641 | 5137 | 999 |

**Table 3.** The impact of the number of IASI pixels (25 km spatial window).

| No. IASI pixels per AMoN–IASI pair | [0, 10) | [10, 20) | [20, 40) | [40, ∞) |
|---|---|---|---|---|
| $r$ | 0.16 | 0.37 | 0.50 | 0.63 |
| Mean temporal coverage per AMoN–IASI pair (%) | 22 | 42 | 61 | 80 |
| No. AMoN–IASI pairs | 4533 | 5025 | 5309 | 676 |

cases where some days have no satellite measurements due to clouds or low thermal contrast. For example, Fig. 1b has 1 missing day ($N = 23$ satellite measurements) but with an otherwise even distribution throughout the remainder of the period, while Fig. 1c ($N = 24$) has nearly the same number of satellite measurements as Fig. 1b but clustered on only 8 of the 15 d. Finally, there are also many cases where selected day(s) have few or no IASI measurements at all (Fig. 1d). When neither temporal coverage nor the number of IASI pixels is high, one can still calculate the matched IASI $NH_3$ column for this AMoN sample, but the result is unlikely to be as representative as a more temporally distributed comparison.

To this end, we explore the dependence of the correlation between IASI and AMoN on IASI data's temporal coverage of the 2-week sampling period and total number of IASI pixels within the 2-week AMoN sampling period, using the 25 km spatial window. For example, the temporal coverages for Fig. 1 are 100 %, 92 %, 53 %, and 14 % and the number of IASI pixels are 52, 23, 24, and 5, respectively. The impact of different temporal averaging and the number of IASI pixel requirements are summarized in Tables 2 and 3, respectively. Increasing temporal coverage and the number of IASI pixels both yield higher $r$ values than any of the simple spatial windows alone. Table 2 shows that the correlation improves to $r = 0.74$ when the temporal coverage is $\geq 80\%$, suggesting a significant impact of temporal coverage of the IASI data. The IASI and AMoN correlations also increase over a simple spatial window with increasing numbers of IASI pixels, yet the impact is not as strong ($r = 0.63$ for $N \geq 40$) as the sensitivity to temporal coverage.

Because the temporal coverage and number of IASI pixels are not independent variables, additional analyses are conducted to study the sensitivity of these two effects using the Monte Carol method. First, the available dataset is filtered to cases when at least 1 of the 14 d have multiple IASI measurements per AMoN measurement, at least 7 d of the 14 d sampling period had at least one IASI measurement, and the total number of IASI pixels is at least 20. The number of days with

available IASI measurements is denoted by $T$. Two opposite approaches are explored for 104 qualified AMoN sites:

1. For maximized temporal coverage (TC_max), only one IASI pixel is randomly selected to represent that day, and the total number of IASI pixels equals $T$ ($T \leq 14$). In this case, the temporal coverage is maximized.

2. For minimized temporal coverage (TC_min), only days with the largest number of IASI pixels are selected until the total number of IASI pixels equals $T$ ($T \leq 14$). In this case, the temporal coverage is minimized, and the total number of selected IASI pixels is the same as TC_max.

For each AMoN site, we repeated the two different sampling strategies 100 times and then calculated the median $r$ value to represent each site using the maximum and minimum coverage approaches. Figure 2a shows the histogram and normalized fit of change in $r$ ($\Delta r = $ TC_max-TC_min) for each site between the two scenarios with the number of bins determined by Sturges' rule. The increased correlation of $\Delta r = 0.45 \pm 0.28$ shows the large impact of temporal coverage. The total number of IASI pixels used for the two strategies was identical.

To further investigate the impact of including more IASI pixels after maximizing temporal coverage, we also test the process described in (1) and then randomly added (20-T) more IASI pixels from the remaining IASI pixels and referred to it as TC_max_add. Figure 2b shows that the changes $\Delta r$ between TC_max and TC_max_add are small ($-0.00 \pm 0.05$). For the TC_max strategy, the initial number of IASI pixels was between 7 and 14, which means that using the TC_max_add strategy results in a $43 \sim 186\%$ increase in the number of IASI pixels compared to TC_max alone. Adding more IASI pixels does not have a significant impact on the $r$ values, indicating that maximized temporal coverage alone is the most important factor when comparing IASI to AMoN stations.

**Table 4.** AMoN and IASI comparison results for different spatial windows (temporal coverage $\geq 80\%$).

| Spatial window | 15 km | 25 km | 50 km | 100 km |
| --- | --- | --- | --- | --- |
| Pearson's $r$ | 0.76 | 0.74 | 0.58 | 0.48 |
| Mean no. IASI pixels per AMoN–IASI pair | 19 | 38 | 119 | 392 |
| No. AMoN–IASI pairs | 105 | 999 | 3138 | 6899 |

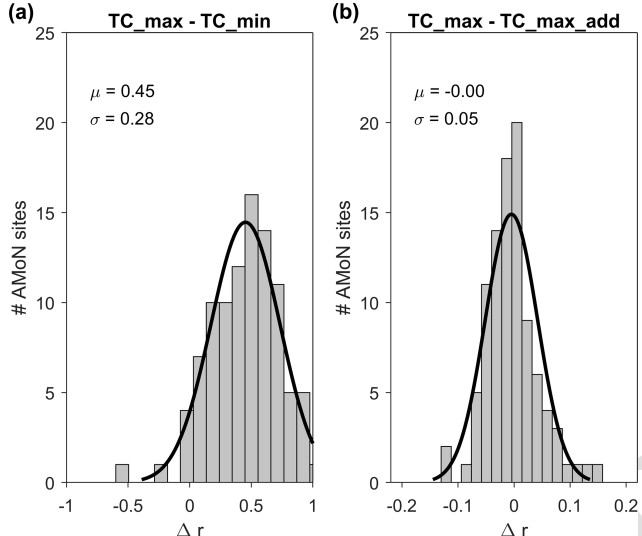

**Figure 2.** The change in $r$ values for individual AMoN sites using different sampling strategies: **(a)** maximized temporal coverage (TC_max) and minimized temporal coverage (TC_min) and **(b)** maximized temporal coverage and randomly adding more pixels (TC_max_add).

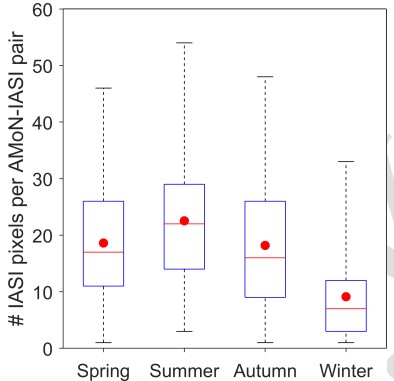

**Figure 3.** Boxplot of number of IASI pixels per AMoN–IASI pair for spring, summer, autumn, and winter. The boxes denote the 25th and 75th percentiles, the whiskers denote the 1st and 99th percentiles, and the red dot denotes the mean.

After applying a temporal coverage requirement (temporal coverage $\geq 80\%$) to filter the overall dataset, we revisit the sensitivity of the agreement between spatial windows. The smaller spatial window now yields better agreement than the larger spatial windows (Table 4). Compared with Table 1, which has no filter for temporal coverage, the $r$ values in Table 4 increase for all spatial windows. The correlations are clearly better for smaller spatial windows ($r = 0.74$ for 25 km versus $r = 0.48$ for 100 km). In this way, the use of a larger spatial window is indeed a tradeoff between the increasing temporal coverage versus incorporating a larger spatial gradient. The results further demonstrate that the IASI pixels far from the AMoN sites may not be representative of the AMoN site.

### 3.3 Sensitivity to seasons and temporal averaging

AMoN has similar numbers of measurements in spring (March, April, May), summer (June, July, August), autumn (September, October, November), and winter (December, January, February), while the mean number of IASI pixels (no. IASI pixels) per pair in winter is only around half that of the other seasons (Fig. 3). In winter, low thermal contrasts result in a low sensitivity of the thermal infrared sounder, which explains the low number of IASI pixels in winter (Clarisse et al., 2010; Guo et al., 2021). The lower sensitivity of the infrared thermal sounder measurements in winter results in higher uncertainties, and thus comparisons between IASI and AMoN are especially important. When temporal coverage is at least 80%, IASI wintertime data still have good agreement with AMoN ($r = 0.61$), although the comparison is limited to only a few AMoN and IASI pairs ($N = 33$). The $r$ values for spring, summer, and autumn when temporal coverage is $\geq 80\%$ are 0.60 ($N = 181$), 0.76 ($N = 502$), and 0.70 ($N = 283$), respectively. IASI in general only provides a small number of pixels in winter; however, it indeed has the capability of reflecting surface $NH_3$ variations even in winter.

The results in Sect. 3.1 TS2 and 3.2 have already shown the importance of spatial window and temporal coverage. The temporal averaging and regridding approaches, such as the tessellation oversampling and physical oversampling, are common methods to achieve higher spatial resolution by sacrificing the temporal resolution (Sun et al., 2018; Van Damme et al., 2018; Wang et al., 2021). Here we neglect the interannual variability in $NH_3$ seasonality and calculate averaged IASI and AMoN $NH_3$ seasonality during 2008–2018 using the 25 km spatial window. By averaging the multiyear IASI data, the impacts of temporal coverage are alleviated because both temporal coverage and number of IASI pixels

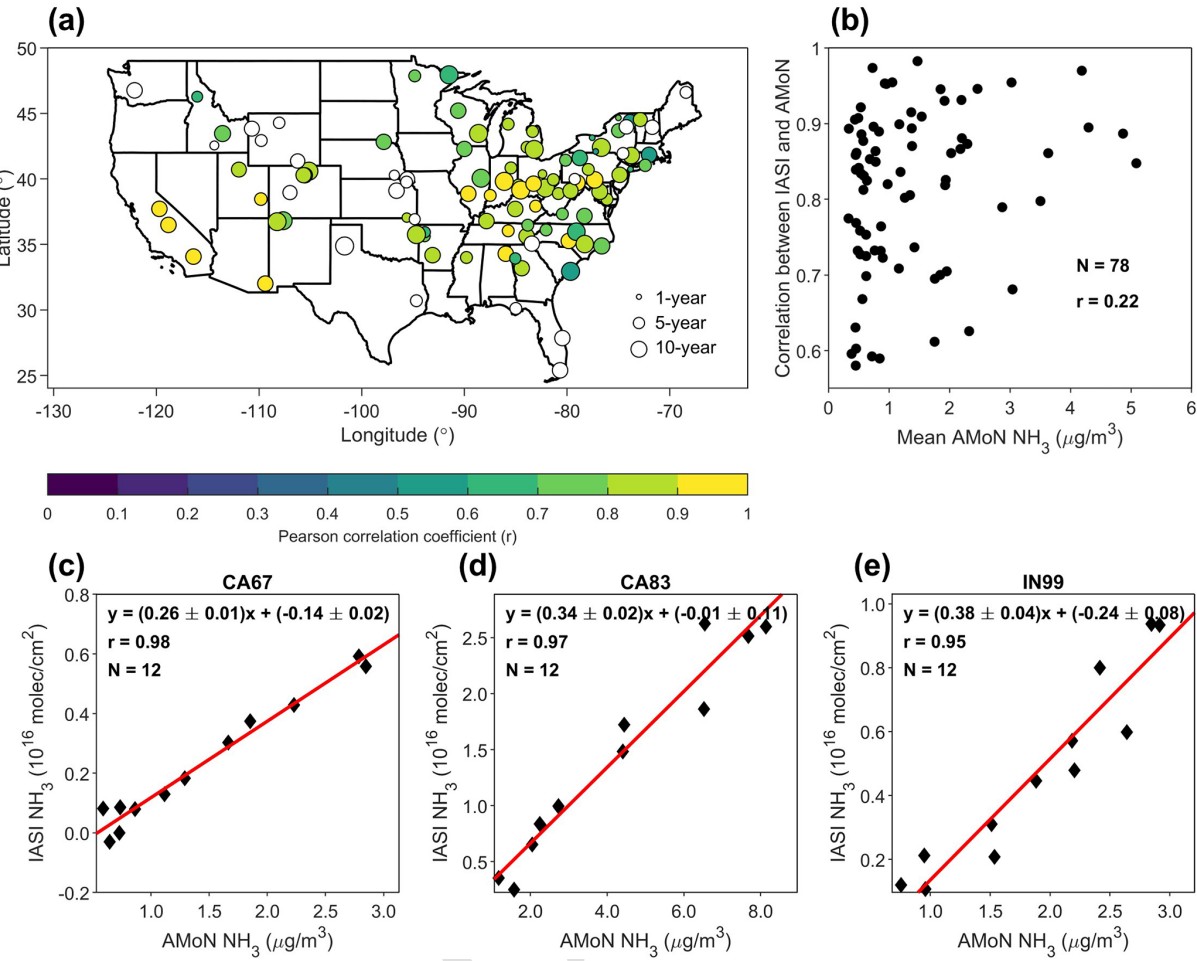

**Figure 4. (a)** Multiyear-averaged NH$_3$ seasonality comparison results between AMoN sites and the IASI observations within 25 km of the AMoN sites at monthly resolution. Circles without filled color denote the AMoN sites with no statistically significant correlation with IASI ($\alpha = 0.05$). The circle sizes denote the length of AMoN data record. **(b)** The relationship between mean AMoN NH$_3$ concentrations and the correlation between AMoN and IASI seasonality. The regression between IASI- and AMoN-observed NH$_3$ seasonality for **(c)** the AMoN site in Joshua Tree National Park, California (CA67); **(d)** the AMoN site in Sequoia National Park, California; and **(e)** the AMoN site in Indianapolis, Indiana (IN99).

increase. Among the 101 AMoN sites with at least 1 full year of data and available IASI v3.1r NH$_3$ data, 49 sites show strong agreement with IASI with $r > 0.8$, 29 sites have moderate agreement of $0.5 < r \leq 0.8$, and 23 sites do not have
5 statistically significant agreements (Fig. 4a). If taking all data into consideration, the overall $r$ value for the CONUS is 0.69. The AMoN sites with higher NH$_3$ concentrations tend to show better agreements between AMoN and IASI (Fig. 4b). The median AMoN NH$_3$ annual mean concentrations for all
10 sites is 0.86 µg m$^{-3}$. Most sites with no statistically significant agreements have a low NH$_3$ concentration (median: 0.48 µg m$^{-3}$). Currently, most AMoN sites are located in regions with low or moderate NH$_3$ concentrations with a lack of sites in the NH$_3$ hotspots (Wang et al., 2021) and urban ar-
15 eas, complicating the comparison between AMoN and IASI.

The above agreement demonstrates that the IASI NH$_3$ column reflects the variation in the surface NH$_3$ concentration at seasonal resolution. For regions without any available ground measurements, IASI NH$_3$ observations can be used to help us better understand the NH$_3$ variations. However, 20 large differences exist among the relationships between IASI and AMoN NH$_3$ concentrations over different AMoN sites (an example of linear regression plot in Fig. 5b). Even for AMoN sites with excellent correlation ($r > 0.8$), the slopes vary a lot, ranging from 0.08–1.4 × 10$^{16}$ molec. cm$^{-2}$ per mi- 25 crogram per cubic meter. For instance, two AMoN sites in California, Joshua Tree National Park (CA 67) and Sequoia and Kings Canyon National Park (CA 83), both exhibit great seasonality agreements with IASI ($r = 0.97$ and $r = 0.99$, respectively), but the slope for CA 83 (Fig. 4d) is 44 % higher 30 than CA 67 (Fig. 4c) TS3. The difference between the slopes

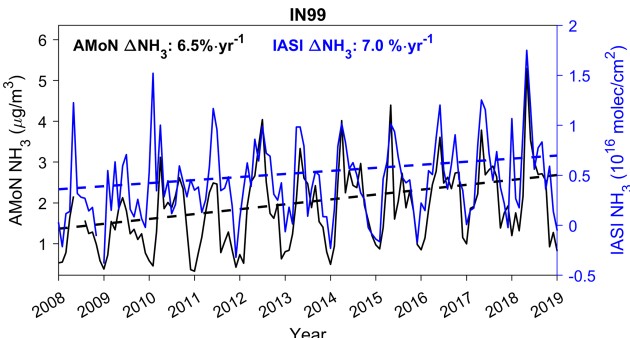

**Figure 5.** The 2008–2018 trends in monthly averaged $NH_3$ for the AMoN site in Indianapolis, Indiana, US (IN 99), and IASI $NH_3$ observations within 25 km of IN 99.

suggests that although IASI is able to capture the general seasonality, the relationship between $NH_3$ column and surface $NH_3$ is distinctly different due to complicated topography, meteorology, and other factors at different AMoN sites.

## 4   Trend analysis

### 4.1   Trend in the CONUS

Strong evidence of increasing $NH_3$ concentrations in the US comes from both ground-based observations and satellite measurements (Van Damme et al., 2021; Warner et al., 2017; Yao and Zhang, 2016; Yao and Zhang, 2019; Yu et al., 2018). The methodology and comparison results in Sect. 3 demonstrate that IASI $NH_3$ can be used to verify and augment regional $NH_3$ trends over the last decade. Figure 5 shows monthly averaged IASI and AMoN time series from Indianapolis, Indiana, USA (IN 99). The strong correlation ($r = 0.96$) between the two measurements is shown in Fig. 5b. Although the $NH_3$ seasonality remains consistent from 2008 to 2018 – namely spring maxima and secondary maxima in autumn with the lowest values in winter – both AMoN and IASI also show increasing trends of $NH_3$ concentrations over the entire time series. AMoN shows a trend of 6.5 % yr$^{-1}$, while IASI shows a trend of 7.0 % yr$^{-1}$.

    Here we will compare IASI $NH_3$ trends with the AMoN-observed $NH_3$ trends in the CONUS over the last decade. We include AMoN trend analysis only for sites with full-year coverage during 2008–2018 ($N = 13$). To achieve a higher spatial resolution, in the following study, we used the oversampled IASI $NH_3$ maps to calculate the $NH_3$ trend for each 2 km grid box. A long-term trend analysis was then performed using AMoN and IASI oversampled data (Sun et al., 2018; Wang et al., 2021) by Theil–Sen's slope estimator and the MK test to examine the agreement between the datasets and explore any regional differences. IASI $NH_3$ columns smaller than the 5th percentile ($0.5 \times 10^{15}$ molec. cm$^{-2}$) of the 11-year $NH_3$ average in the CONUS region were excluded to avoid spurious trend results caused by the higher

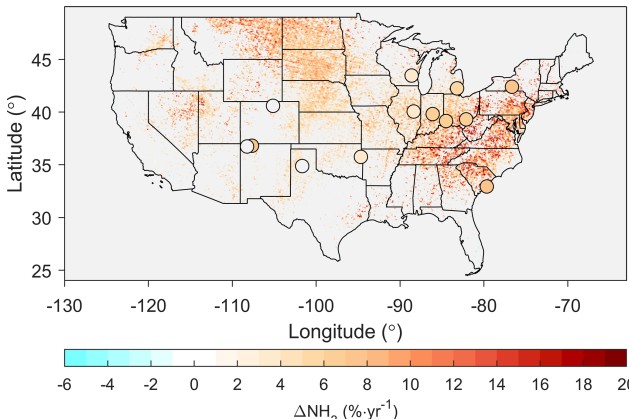

**Figure 6.** Trend analysis for IASI $NH_3$ (2008–2018) and AMoN $NH_3$ measurements in the contiguous US. The gray color indicates no statistically significant change ($\alpha = 0.05$).

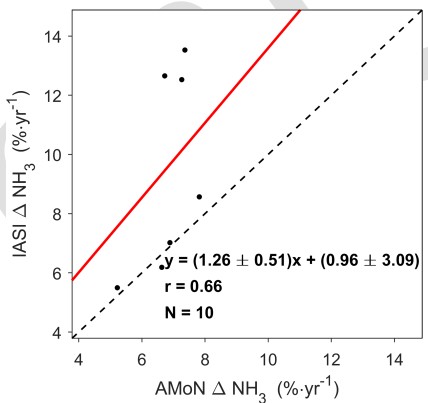

**Figure 7.** Comparison between 2008–2018 AMoN and IASI $NH_3$ trends (25 km spatial window) for AMoN sites with available nearby IASI trend data.

noise in these measurements. To perform the interannual trend analysis, we require each region or site to have at least one valid measurement in each season to alleviate the possible bias due to seasonal variations. Figure 6 shows the annual percentage change for both IASI and AMoN. Most regions in the CONUS have increasing $NH_3$ concentrations based on the 11-year IASI observations (median: 6.8 % yr$^{-1}$), including the eastern US, Midwest, and parts of the western US. A total of 10 out of 13 AMoN sites have statistically significant $NH_3$ increases. AMoN data in general suggest similar increases (median: 6.7 % yr$^{-1}$). When plotting the trends of AMoN sites against the median of IASI trends within a 25 km spatial window (Fig. 7), a moderate correlation ($r = 0.66$) was found between IASI and AMoN $NH_3$ trends. IASI in general suggested a higher $NH_3$ increase compared to AMoN (slope: $1.26 \pm 0.51$), with the ratio larger than 1 for most sites.

The spatial consistency across the datasets differs significantly. Both AMoN and IASI suggest $\sim 5\,\%\,\mathrm{yr}^{-1}$ NH$_3$ increases in the Great Lakes region, while IASI suggests a higher NH$_3$ increase in the eastern US compared with AMoN. The IASI trend analysis results suggest a significant NH$_3$ increase in the northern Great Plains, e.g., North Dakota, South Dakota, and Montana, yet there are no AMoN sites in this region. Furthermore, the trends are consistent with the NH$_3$ emissions increases caused by increased N fertilizer usage in the northern Great Plains (P. Cao et al., 2020). McHale et al. (2021) showed that wet-precipitation NH$_4^+$ concentrations based on NADP observations suggested the highest increases in the Great Plains, the Rocky Mountain region, and the Great Lakes region from 2000 to 2017, which is geographically consistent with the NH$_3$ trends observed by both AMoN and IASI. Here we note that the spatial resolution could affect the results of trend analyses. The trend $6.8\,\%\,\mathrm{yr}^{-1}$ was derived as the median of trends for each 2 km grid box. If considering the CONUS as a whole and calculating the annual mean NH$_3$ for the whole CONUS during 2008–2018 to derive the overall trend in the CONUS, the IASI NH$_3$ change for 2008–2018 is $(3.9 \pm 2.2)\,\%\,\mathrm{yr}^{-1}$ and $(1.3 \pm 0.8) \times 10^{14}\,\mathrm{molec.\,cm}^{-2}\,\mathrm{yr}^{-1}$, similar to the trend in the previous study of $(3.4 \pm 0.6)\,\%\,\mathrm{yr}^{-1}$ and $(1.1 \pm 0.4) \times 10^{14}\,\mathrm{molec.\,cm}^{-2}\,\mathrm{yr}^{-1}$ (Van Damme et al., 2021).

We use the Hoshen–Kopelman algorithm to cluster adjacent grid points above the 95th percentile threshold of the 11-year CONUS oversampling map ($6.7 \times 10^{15}\,\mathrm{molec.\,cm}^{-2}$) as a NH$_3$ hotspot (Hoshen and Kopelman, 1976; Wang et al., 2021), and the median area of identified hotspots is $\sim 150\,\mathrm{km}^2$ (Wang et al., 2021). Analyzing NH$_3$ hotspots, the median of the NH$_3$ trend is $4.7\,\%\,\mathrm{yr}^{-1}$, indicating that the regions of the largest emissions sources are also seeing increasing concentrations over time. Although the percent changes in the regions with the highest concentrations are smaller than the trend in the CONUS median ($6.8\,\%\,\mathrm{yr}^{-1}$), in terms of the absolute changes, the median trends of NH$_3$ columns over these NH$_3$ hotspots are higher than the trend in the CONUS median ($3.7 \times 10^{14}$ versus $2.8 \times 10^{14}\,\mathrm{molec.\,cm}^{-2}\,\mathrm{yr}^{-1}$). The top 10 NH$_3$ hotspots in the CONUS regarding column–areal weighting (NH$_3$ column times the area) all exhibit increasing NH$_3$ concentrations from 2008 to 2018 (Table 5). Within these hotspots, the central Great Plains experience the largest NH$_3$ increase (median: $5.0\,\%\,\mathrm{yr}^{-1}$, $4.0 \times 10^{14}\,\mathrm{molec.\,cm}^{-2}\,\mathrm{yr}^{-1}$), while the San Joaquin Valley (median: $2.0\,\%\,\mathrm{yr}^{-1}$, $1.6 \times 10^{14}\,\mathrm{molec.\,cm}^{-2}\,\mathrm{yr}^{-1}$) and Imperial County, California (median: $2.1\,\%\,\mathrm{yr}^{-1}$, $1.9 \times 10^{14}\,\mathrm{molec.\,cm}^{-2}\,\mathrm{yr}^{-1}$), see the smallest changes.

To provide a detailed insight of the increasing NH$_3$ over the CONUS, we further perform trend analyses for different seasons (Fig. 8). In spring, significant NH$_3$ increases are found in the Midwest and in the eastern US. In summer, NH$_3$ increases shift to the western US and the northeast US.

**Table 5.** The 2008–2018 IASI-observed NH$_3$ trend in the top 10 NH$_3$ hotspots (column–areal weighting) in the CONUS.

| Hotspots | % yr$^{-1}$ | $10^{14}\,\mathrm{molec.\,cm}^{-2}\,\mathrm{yr}^{-1}$ |
|---|---|---|
| Central Great Plains | 5.0 | 4.0 |
| The San Joaquin Valley | 2.0 | 1.6 |
| North Oklahoma | 3.9 | 2.9 |
| Texas panhandle | 3.6 | 2.8 |
| Central Iowa | 4.4 | 3.3 |
| The Snake River Valley | 3.8 | 3.3 |
| Southeast Iowa | 5.2 | 3.9 |
| Beadle County, South Dakota | 8.3 | 6.0 |
| Weld County, Colorado | 3.6 | 2.9 |
| Imperial County, California | 2.1 | 1.9 |

AMoN and IASI seasonality clustering results show that the Midwest and eastern US, dominated by fertilizer NH$_3$ emissions, have a broad spring maximum of NH$_3$, while the western United States, dominated by volatilization of livestock waste NH$_3$ emissions, in contrast, shows a narrower midsummer peak (Wang et al., 2021). The spatial patterns of spring and summer NH$_3$ trends are in agreement with the seasonality clustering results, indicating that increasing NH$_3$ emissions caused by agricultural activities may contribute to an NH$_3$ concentration increase. The increasing wildfire activities in the western US may also contribute to NH$_3$ increases (Lindaas et al., 2021a, b). In autumn and winter, most regions in the US do not have statistically significant IASI NH$_3$ trends, and a decreasing NH$_3$ trend is observed by IASI in the southwest US in autumn. In contrast, AMoN data suggest a notable NH$_3$ increase in the northeast and the Corn Belt region in winter. Again, IASI data are susceptible to low thermal contrasts in winter, which to some extent explains the disagreement between IASI and AMoN in winter, as discussed in Sect. 3.3.

Wintertime NH$_3$ plays an important role in haze episodes through the formation of aerosol-phase NH$_4$NO$_3$ (Shah et al., 2018; Zhai et al., 2021), and increasing NH$_3$ concentrations in winter may affect aerosol acidity and aerosol chemistry (Lawal et al., 2018; Zheng et al., 2020). In the past decades, NO$_x$ and SO$_2$ emissions reductions have resulted in less NH$_x$ partitioning into particle-phase NH$_4^+$ (Shah et al., 2018); however, the partitioning alone is not able to fully explain the significant NH$_3$ concentration increases (Yao and Zhang, 2019; Yu et al., 2018). The change in meteorological conditions, such as increasing air temperatures, may also contribute to the increasing NH$_3$ trends (Warner et al., 2017; Yao and Zhang, 2019). No matter the reason for increasing NH$_3$ concentrations across the CONUS regions, the fact that both NH$_3$ surface concentrations and NH$_3$ column concentrations have been increasing during the past decade will have significant impacts on air quality and nitrogen deposition. EPA is reviewing the 2020 PM$_{2.5}$ National Ambient Air Quality Standard (NAAQS) currently set at $12.0\,\mu\mathrm{g\,m}^{-3}$, and if the NAAQS is lowered, NH$_3$ controls will become in-

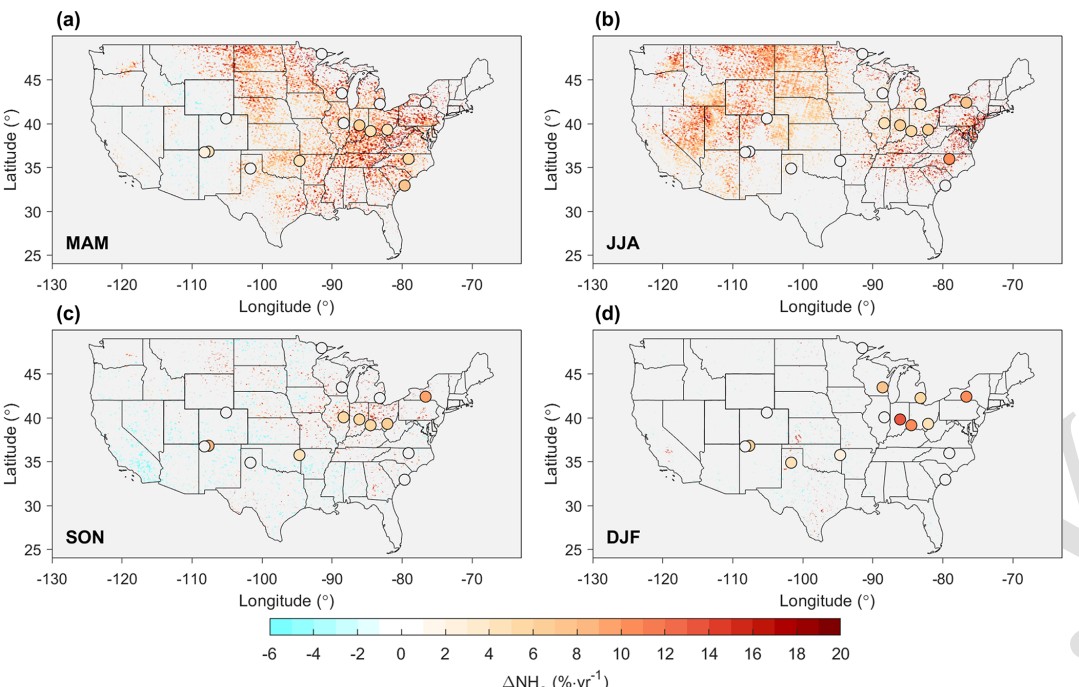

**Figure 8.** The 2008–2018 NH$_3$ trend for different seasons based on IASI NH$_3$ measurements in the contiguous US. **(a)** Spring (March, April, May); **(b)** summer (June, July, August); **(c)** autumn (September, October, November); **(d)** winter (December, January, February). The gray color indicates no statistically significant change ($\alpha = 0.05$).

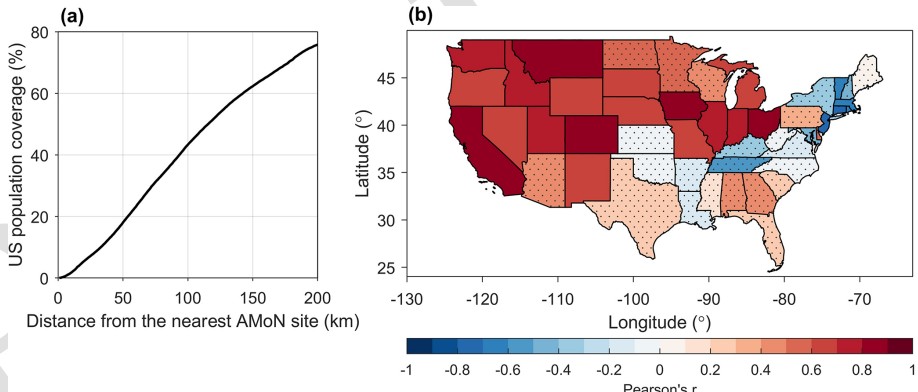

**Figure 9.** **(a)** Cumulative distribution of the CONUS population as a function of distance from the nearest AMoN site; **(b)** correlation between EPA NH$_3$ emissions and IASI-observed mean NH$_3$ concentrations at state level during 2008–2018. The gray dots represent states without statistically significant correlations ($\alpha = 0.05$).

creasingly important for meeting the standard. Additionally, Pan et al. (2021) demonstrate that NH$_3$ transported from Colorado significantly increased the dry NH$_3$ deposition in the Rocky Mountain National Park. Increasing gas-phase NH$_3$ may result in longer spatiotemporal scales for dry nitrogen deposition, leading to adverse impacts on remote regions and sensitive ecosystems (Phoenix et al., 2006). A reduction in NH$_3$ emissions is critical to protect human health and the biodiversity in sensitive ecosystems (Ellis et al., 2013; Hill et al., 2019).

## 4.2 Trend in the urbanized areas

The short lifetime of NH$_3$ leads to strong spatial variabilities in NH$_3$ concentrations, and most AMoN sites are not located in highly populated urban regions (Wang et al., 2021), a gap that IASI data can fill. Figure 9a shows the cumulative distribution of the US population as a function of the distance from an AMoN site. Population data were retrieved from the Gridded Population of the World, Version 4 (GPWv4) (Center for International Earth Science Information Network – Columbia University, 2018). More than half

**Table 6.** The 2008–2018 IASI NH$_3$ trend in the top 10 most populous urbanized areas.

| Urbanized area | Population (million) | % yr$^{-1}$ | 10$^{14}$ molec. cm$^{-2}$ yr$^{-1}$ TS5 |
|---|---|---|---|
| New York–Newark, NY–NJ–CT | 18.0 | 10.8 | 2.0 |
| Los Angeles–Long Beach–Anaheim, CA | 12.0 | 4.3 | 2.1 |
| Chicago, IL–IN | 8.6 | 5.2 | 2.5 |
| Miami, FL | 5.5 | −25.2 | −1.5 |
| Philadelphia, PA–NJ–DE–MD | 5.4 | 10.9 | 2.6 |
| Dallas–Fort Worth–Arlington, TX | 5.1 | – | – |
| Houston, TX | 4.9 | 7.9 | 2.0 |
| Washington, DC–VA–MD | 4.6 | 9.0 | 2.2 |
| Atlanta, GA | 4.5 | 9.4 | 2.2 |
| Boston, MA–NH–RI | 4.2 | 10.5 | 1.4 |

of the CONUS population is at least 100 km away from an AMoN site. As mentioned in the previous discussion of spatial windows, AMoN may best represent the NH$_3$ variations for regions within a $\sim$ 10 km radius, and less than 2 % of the CONUS population is within 10 km of an AMoN site. More urban AMoN sites are needed to represent the urban areas and better quantify NH$_3$ emissions from mobile sources and trends in population centers. Satellite observations are the only dataset that can currently be used to investigate source contributions and trends in population centers (Cao et al., 2022).

We retrieved urban area data from the 2010 US Census, which includes two different types of urban areas: urbanized areas (UAs) of 50 000 or more people and urban clusters (UCs) of at least 2500 and less than 50 000 people (US Census Bureau, 2012). The urban areas have a similar NH$_3$ trend compared with the CONUS (8.1 versus 6.8 % yr$^{-1}$), suggesting a simultaneous NH$_3$ increase in both urban and rural areas. The top 10 most populous urbanized areas almost all exhibit significant NH$_3$ increases with the exception of Miami, Florida, which has a negative trend, and Dallas, Texas, without any significant trend (Table 6). These 10 areas in total account for more than 70 million people, making up more than one-fifth of the total population in the CONUS. The urban environment with abundant HNO$_3$ and NH$_3$ emissions from vehicles favors the formation of NH$_4$NO$_3$. Recent studies suggest that gas-phase NH$_3$ hinders the scavenging of NH$_4$NO$_3$ by slowing down the deposition process of total inorganic nitrate (Zhai et al., 2021) and promotes new atmospheric particle formation by directly nucleating with HNO$_3$ to form NH$_4$NO$_3$ in winter in urban areas (Wang et al., 2020). However, ultimately the sensitivity to PM$_{2.5}$ from increases in NH$_3$ in any urban area will be a complex function of trends of NO$_x$ and SO$_2$ as well (Feng et al., 2020). The NH$_3$ increase in these densely populated areas and its impact on aerosol chemistry need to be further addressed. For example, Fig. 9b shows the relationship between NH$_3$ trends versus emissions trends (EPA Air Pollutant Emissions Trends Data, 2023) on the state level. For agricultural areas

with high NH$_3$ (excess NH$_3$ relative to NH$_4$NO$_3$ equilibrium), one would expect an increase in emissions to correlate very well with increasing NH$_3$ columns. In contrast, in areas with more NO$_x$, increases in emissions may result in NH$_3$ going into NH$_4$NO$_3$ and thereby show little or even negative correlations. To this end, Fig. 9b shows that at state level, states with strong agricultural emissions show strong correlations between emissions and concentration trends, e.g., Iowa, while northeast states show weak or negative correlations, e.g., New Jersey. Ultimately, co-located aerosol-phase and gas-phase precursor measurements are needed to fully deduce what is happening at each urban area and should be a focus of future air quality network integration.

## 5 Implications

Under favorable conditions, IASI NH$_3$ columns correlate with AMoN NH$_3$ surface concentrations even at the 2-week scale and for low-concentration regions ($r = 0.74$ when temporal coverage $\geq$ 80 %). The temporal coverage of IASI data during the 2-week AMoN sampling period is the controlling factor of the correlation between IASI and AMoN measurements, presumably because of the large day-to-day variability in NH$_3$. The agreement demonstrates the strong potential for using IASI NH$_3$ columns to bridge the spatial gaps of the AMoN network. The global coverage of satellite measurements enables the IASI NH$_3$ product to serve as an alternative dataset in countries and regions that do not have any NH$_3$ monitoring networks, particularly in developing countries. For example, India is the second most populated country in the world with a sixth of the world's population, and a recent study has shown the unique role of NH$_3$ in forming massive chloride aerosols (up to 40 µg m$^{-3}$) in India (Gunthe et al., 2021). However, there are currently no long-term NH$_3$ ground monitoring networks in India, impeding the efforts to estimate and control NH$_3$ emissions (Beale et al., 2022). IASI's low sensitivity to wintertime NH$_3$ shows the value of the more sensitive AMoN sites. Extra attention is needed when using IASI data in such circumstances.

The increasing $NH_3$ in the CONUS (median: $6.8\,\% \, yr^{-1}$, $2.8 \times 10^{14}$ molec. $cm^{-2} \, yr^{-1}$), including the hotspot regions (median: $4.7\,\% \, yr^{-1}$, $3.7 \times 10^{14}$ molec. $cm^{-2} \, yr^{-1}$), highlights the more important role of $NH_3$ in $PM_{2.5}$ formation and nitrogen deposition in the future. AMoN suggests a similar $NH_3$ increase ($6.7\,\% \, yr^{-1}$), as well as similar spatial patterns with IASI. Both IASI and AMoN show the largest $NH_3$ increases in the Midwest and eastern US, with a moderate correlation between the IASI and AMoN trends for the entire CONUS ($r = 0.66$). More co-located measurements of $PM_{2.5}$ mass and $NH_3$ concentrations would help in assessing the impact that increasing trends of $NH_3$ will have on human health. The integrated satellite and ground-based measurements are already playing a role in our understanding of under-represented $NH_3$ emissions sources in the inventories. $NH_3$ already dominates the reactive nitrogen deposition in most regions in the US; because of the continuing efforts on $NO_x$ emission reductions, $NH_3$ is expected to become the key species for nitrogen deposition (Li et al., 2016), which will have adverse impacts on the nearby ecosystem regions, e.g., the national parks (Benedict et al., 2013; Pan et al., 2021). The changing partitioning of $NH_x$ between $NH_3$ and $NH_4^+$ is likely to impact the lifetime of $NH_x$ due to differences between the removal velocity of gas-phase $NH_3$ via dry deposition and particle-phase $NH_4^+$ wet deposition. The trends vary in different seasons, with $NH_3$ increases mainly in spring in the Midwest and eastern US (cropland dominated) and in summer in the western US (feedlot dominated), suggesting the impacts from agricultural activities and the necessity of developing regionally specific emission control strategies.

Because of the scarcity of ground monitoring sites in the urban areas, satellite $NH_3$ measurements are extremely valuable for characterizing $NH_3$ magnitude, seasonality, and trends in densely populated areas. Satellite observations suggests $NH_3$ increases across the US urban areas (median: $8.1\,\%$). New York–Newark, NY–NJ–CT, alone has a population of more than 18 million, experiencing a $10.8\,\% \, yr^{-1}$ $NH_3$ increase. Measurements from satellites will help inform where ground-based $NH_3$ samplers could be located to better understand local air quality in overburdened communities with limited resources for continuous monitors. In addition, $NH_3$ sources in the urban areas and the related atmospheric chemistry are both poorly understood (Gu et al., 2022; Sun et al., 2017) and could be constrained by satellite $NH_3$ observations (Cao et al., 2022). However, satellite observations alone are not able to answer all questions under the complex urban atmospheric conditions. For instance, gas-phase $NH_3$ and $HNO_3$ can nucleate directly to form $NH_4NO_3$ particles in cold atmospheric conditions and is likely to result in the rapid growth of new atmospheric particles in winter in urban areas (Wang et al., 2020). The comparison between $NH_3$ emission trends and IASI-observed $NH_3$ concentration trends suggests that strong correlations exist in states with large $NH_3$ emissions from agricultural activities, e.g., Iowa, while there are weak or negative correlations in northeast states, e.g., New Jersey, indicating the different contribution from emissions and partitioning. To provide accurate and fine-spatial-scale $NH_3$ observations in the urban areas, more routine ground monitoring sites are needed in both urban areas and high-$NH_3$ emission source regions.

**Data availability.** The AMoN data were downloaded from the National Atmospheric Deposition Program/National Trends Network (NADP/NTN): https://nadp.slh.wisc.edu/networks/ammonia-monitoring-network/ (NADP, 2023). The authors acknowledge the AERIS data infrastructure (https://www.aeris-data.fr, AERIS Data and Services for the Atmosphere, 2022) for providing access to the IASI Level 2 $NH_3$ data used in this study. Population data were retrieved from the Center for International Earth Science Information Network, Columbia University: https://sedac.ciesin.columbia.edu/data/collection/gpw-v4/ (Center for International Earth Science Information Network, Columbia University, 2018). The urban areas data are downloaded from the US Census Bureau: https://www.census.gov/geographies/mapping-files.html (US Census Bureau, 2012). The emission trend data are downloaded from the US Environmental Protection Agency's Air Pollutant Emissions Trends Data: https://www.epa.gov/air-emissions-inventories/air-pollutant-emissions-trends-data (EPA, 2023b). TS6

**Author contributions.** MAZ and RW designed the research; RW led the analysis; KS, DP, and XG contributed to data analysis; LC, MVD, PFC, and CC helped with the usage of IASI data; MP helped with the usage of AMoN data; and RW wrote the paper with contributions from all co-authors.

**Competing interests.** The contact author has declared that none of the authors has any competing interests.

**Disclaimer.** The research presented was not performed or funded by EPA and was not subject to EPA's quality system requirements. The views expressed in this article are those of the author(s) and do not necessarily represent the views or the policies of the US Environmental Protection Agency.

Publisher's note: Copernicus Publications remains neutral with regard to jurisdictional claims in published maps and institutional affiliations. TS7

**Acknowledgements.** Mark A. Zondlo and Rui Wang would like to gratefully acknowledge support from the NASA Health and Air quality Applied Sciences (HAQAST) team. Mark A. Zondlo acknowledges support as a visiting scientist at ULB from the EU-METSAT Satellite Application Facility on Atmospheric Chemistry Monitoring. Xuehui Guo gratefully acknowledges the support from the NASA Earth and Space Science Fellowship. Kang Sun wants to

thank the NASA Atmospheric Composition Modeling and Analysis Program for their support. Cathy Clerbaux is grateful to CNES for scientific collaboration. The authors acknowledge the AERIS data infrastructure for providing access to the IASI data in this study. CE1

**Financial support.** This research has been supported by the NASA Health and Air Quality Applied Sciences (HAQAST) team (grant no. NNX16AQ90G); the NASA Earth and Space Science Fellowship (grant no. 80NSSC17K0377); the NASA Atmospheric Composition Modeling and Analysis Program (grant no. 80NSSC19K0988); the Belgian State Federal Office for Scientific, Technical and Cultural Affairs (Prodex HIRS); and the Air Liquide Foundation TS8 (TAPIR project). This work is also partly supported by the 100 TS9 FED-tWIN project ARENBERG ("Assessing the Reactive Nitrogen Budget and Emissions at Regional and Global Scales") funded via the Belgian Science Policy Office TS10 (BEL-SPO) and CNES TS11. TS12 CE2

**Review statement.** This paper was edited by Leiming Zhang and reviewed by two anonymous referees.

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

## Remarks from the language copy-editor

CE1      Please verify the section.
CE2      Please verify the English language edits made in this section.

## Remarks from the typesetter

TS1      Please check change. Update inserted.
TS2      Please check change/addition due to guidelines.
TS3      Please give an explanation of why this needs to be changed. We have to ask the handling editor for approval. Thanks.
TS4      Please check the unit.
TS5      Please check the unit.
TS6      Please check added citations and confirm the section.
TS7      Please confirm this section.
TS8      Is "Fondation Air Liquide, France" meant here?
TS9      Please check "100".
TS10      Is this the same as "Belgian Federal Science Policy Office"?
TS11      Is this the same as "Centre National d'Etudes Spatiales"?
TS12      Please confirm both acknowledgements and financial support sections. Please also check added funders/grant numbers in the MS records.
TS13      Please check; is there a direct link/DOI?
TS14      Please confirm addition.
TS15      Update inserted. Please confirm.
TS16      Please confirm both EPA (2023)-references and make sure it is distinguished with the labels in the text. Thank you.
TS17      Please confirm addition.
TS18      Please check page range/article number and provide DOI.
TS19      Please confirm addition.
TS20      Please provide ISBN/DOI.
TS21      Please confirm addition.