# Peer review of "Bridging the spatial gaps of the Ammonia Monitoring Network using satellite ammonia measurements"

_EGUsphere, 2023_

## Referee Comment (RC1)

**Review of "Bridging the spatial gaps of the Ammonia Monitoring Network using satellite ammonia measurements"**

**Summary**

Multiple observations, both ground based and from satellites, strongly suggest that NH3 concentrations are increasing; these increases will significantly impact ecosystems, air quality and human health. This paper presents the results of an important study that demonstrates the validity of using NH3 data from IASI instruments to extend the limited information available from surface monitoring networks. This is an important capability, since surface networks are in general sparse, or in many regions, non-existent.

The authors compare IASI NH3 columns with the two week surface means from the North American AMoN network and show that the IASI and AMoN data are mostly well correlated, as long as there is good temporal coverage by IASI during the two week AMoN measurement period and only IASI data within 25 km of the AMoN site are used. They then calculate trends from both datasets and show that they are comparable. Having established that IASI and AMoN data provide similar trends, they use the IASI data to calculate trends across the CONUS region, both annually and seasonally. They find that NH3 is increasing faster than 10%·yr-1 in the eastern U.S. and Midwest in the spring and in the western U.S. in the summer. Trends in NH3 "hotspots" (e.g., central Iowa), and urban areas (e.g. New York), which are mostly far from any AMoN site, are also shown to be positive and significant.

The paper is very well organized and the results are certainly important. There are some sections that could be more clearly written, as I have detailed below. I recommend the paper be published with minor revisions.

**Presentation issues**

**Section 2.2**: the authors state that they used AMoN data for all sites except UT01. This is true for the correlation analysis, but not the trend analysis, which used a much more limited set, as is explained later. This should be made clear in section 2.3.2.

**Section 2.3.1**: were the maps really created using the data for entire 2008-2018 period? Based on the discussion in section 4 it seems that they were created for each month of each year, which would be the only way to generate Figure 5. Or maybe multiple versions of the maps were created and used for different estimates? Please make this clear.

**Section 2.3.2:** the authors state that they use the Mann-Kendall test and the Theil-Sen slope estimator for trend analyses. This is a good approach, but never again do they mention either technique. The reader is left to assume that the Theil-Sen slope estimator is used in the trend analysis section, and it's not clear where the Mann-Kendall test is used at all, since all the correlations are presented as Pearson coefficients. If the Mann-Kendall test and the Theil-Sen

slope estimator are only used in the trend section, please make this clear, and then provide a few sentences demonstrating how they are applied.

**Tables 2 and 3**: it seems that the first row captions have been switched between Tables 2 and 3. And please define pair. Does it mean all the IASI pixels co-located with an AMoN site during one two week period? It would be better to say: AMoN-IASI pair. Or add the following on line 238: … for comparison, establishing an AMoN-IASI pair.

**Section 3.3**

Please provide the correlations between IASI and AMoN for all four seasons and not just winter.

Please clarify how Figure 4 was obtained. The text suggest that IASI data were oversampled over the entire 2008-2018 period. I don't see how correlations can be calculated, since such oversampled would have no temporal information. Was the oversampling done on a month by month basis? This point is key for understanding how the results in section 4 were obtained.

I suggest a scatter plot of the AMoN median values and the Pearson coefficient. This would reinforce the conclusions in the paragraph starting at line 360.

The section in the last paragraph in section 3.3 starting at line 377 (However, …) jumps ahead to discuss results in the next section. I think the authors should move this text to the next section, and use the plot I suggested above here to make the points stated at the beginning of this paragraph.

**Section 4**

Please provide more detail on the calculation of the trends: were the trends calculated on each 2 km grid box, how long were the averaging time periods, how were the IASI trends averaged to compare with AMoN trends, how were the IASI data averaged to provide regional, hotspot and CONUS trends. This information is critical for understanding the results discussed in section 4.1 and 4.2. For example, it's hard to understand how the CONUS trend can be 3.9%/yr (line 428), 8.0%/yr (line 436) and 6.8%/yr (line 528). Either there is an error or these three values are calculated differently, but it's not clear what the difference is.

**Figure 5:** It would be interesting to show this figure for a few more sites; this would provide the reader with an idea of the variability in the slope and correlation; the sites mentioned at the end of section 3 (CA67 and CA 83) should be shown and could be discussed here.

**Figure 6:** are the IASI trends calculated in each 2 km grid box?

What is the spatial extent of each "hotspot" box?

**Minor revisions**

Line 17**:** The limited number of NH3 observations hinders …

Line 19: …networks are few and sparse across most of the globe,

Line 34: (2008-2018), suggesting the $NH_3$ will become a greater contributor to nitrogen deposition. $NH_3$ trends at AMoN sites are correlated with IASI NH3 trends (r=0.6), and show similar spatial patterns, with the highest increases in the Midwest and eastern U.S

Line 37:  respectively. NH3 hotpots (defined as regions where the IASI NH3 column is larger than the 95th percentile of 11-year CONUS map, 6.7 Å~ 1015 molec/cm2),are also experiencing increasing concentrations over time, with a median of NH3 trend of 4.7% · yr-1.

Line 42: The increases in NH3 …

Line 43: areas, and therefore should be carefully monitored and studied.

Line 52: $NH_4NO_3$

Line 55:  emissions are decreasing under new pollution controls

Line 58: in sensitive ecosystems.

Line 79: Move definition of IMPROVE to this line.

Line 85: hotspot regions

Line 94: (S-NPP) and on JPSS-1 and JPSS-2

Line 113: validation efforts were carried out in specific seasons

Line 126: in the atmosphere is limited

Line 146: in more detail.

Line 167: on bi-weekly/seasonal

Line 169: We avoided converting column $NH_3$ into surface concentrations

Line 178: sounder deployed on board

Line 183: The latest version is a reanalysis dataset that uses the European ….

Line 185: Because these meteorological data are coherent in time, the reanalyzed NH3 dataset is the most appropriate one to study trends.

Line 200: imagery, we determined

Line 214: this algorithm weights IASI measurements by their uncertainties, which include varying sensitivities to thermal contrast, as described

Line 218: The sentence : "For each season, we were able to achieve sufficiently overlapped IASI pixels through calculating the sum of the unnormalized spatial response function (SRF) of the oversampling results" is not clear. Please clarify: a bit more detail might be helpful.

Line 229: Unlike simple linear regressions,

Line 236: For the initial analysis, we first used the simplest method for comparing …

Line 237: for each each AMoN site, we average all IASI observations within a given radius of the site during the AMoN sampling time frame (2 weeks)

Line 239: between the two datasets

Line 250: and the number of IASI pixels

Line 254: or in regions

Figure 1: label each plot with the AMoN site location

Line 267: the 2-week AMoN integration period, (using a 25 km spatial window), could affect the results.

Line 276: as representative as

Line 281: for most days

Line 285: To this end, we explore the dependence of the correlation between IASI and AMoN on the IASI data temporal coverage of the 2-week sampling period and total number of IASI pixels within the 2-week AMoN sampling period, using the 25 km spatial window.

Line 288: The impact of temporal coverage and the number of IASI pixels within the sampling period are

Line 314: For each AMoN site, we repeated the two different sampling strategies 100 times

Line 361:  Temporal averaging and regridding approaches, such as tessellation oversampling and physical oversampling, are common methods used to achieve higher spatial resolution

Line 364: neglect the interannual variability and calculate the multi-year averaged IASI NH3 concentrations, both annual and seasonal, using the 25 km

Line 366: coverage and numbers of IASI pixels increase

Line 393: The methodology and comparison results in section 3 demonstrate that IASI NH3 can be used to verify and augment regional NH3 trends over the last decade. Here we will compare IASI NH3 trends with the AMoN observed NH3 trends in the CONUS region over the last decade.

Line 398: from Indianopolis

Figure 5: shouldn't the caption read "2008-2018 trends in monthly averaged NH3"?

Line 417: Remove the sentence starting with "The absolute" as this information is presented in the next paragraph.

Line 433: Analyzing NH3 hotspots,

Line 434: indicating that the regions with the largest emissions are also seeing concentrations increasing with time.

Line 436: smaller that the trend in the CONUS median

Line 437: higher than the trend in the CONUS median

Line 438: define column-areal weighting

Line 441: see the smallest changes.

Line 456: in the eastern US

Line 457: western US and the Northeast

Line 487: add some text like: (Wang et al., 2021), **a gap that IASI data can fill**.

Line 488: the cumulative distribution of the CONUS population as a function of the distance from an AMoN site.

Line 493: mobile sources and trends in population centers.

Line 502: in total account for more than seventy million people

Figure 9: Cumulative distribution of CONUS population as a function of distance from an AMoN site.

Line 517: The temporal coverage of the IASI data during the two week AMoN sampling period is the controlling factor of the correlation between the IASI and AMoN measurements, presumably because of the large day-to-day variability of NH3.

Line 521: the IASI NH3 product

Line 522: shown the unique role

Line 530: as well as similar spatial patterns.

Line 531: show the largest increases in the Midwest and eastern U.S., with a moderate correlation between the IASI and AMoN trends for the entire CONUS .

Line 535: deposition in most regions in the U.S.;

Line 536: (2016), which will have adverse impacts

Line 547: communities with limited resources

Line 540: (cropland dominated) and in summer in the western U.S. (feedlot dominated), highlighting the impacts

Line 543: for characterizing NH3 magnitude

---

## Author Comment (AC1)

Response to Review#1
Title: "Bridging the spatial gaps of the Ammonia Monitoring Network using satellite ammonia measurements"
Authors: R. Wang et al.

We thank the reviewer for the helpful comments, careful review, and valuable suggestions of the entire manuscript. We have revised the manuscript accordingly to help clarify and focus the manuscript. The original comments from reviewers are in *blue and italics*, our replies are in black font, and verbatim responses from the revised manuscript are in red font.

**Summary**

*Multiple observations, both ground based and from satellites, strongly suggest that $NH_3$ concentrations are increasing; these increases will significantly impact ecosystems, air quality and human health. This paper presents the results of an important study that demonstrates the validity of using $NH_3$ data from IASI instruments to extend the limited information available from surface monitoring networks. This is an important capability, since surface networks are in general sparse, or in many regions, non-existent. The authors compare IASI $NH_3$ columns with the two week surface means from the North American AMoN network and show that the IASI and AMoN data are mostly well correlated, as long as there is good temporal coverage by IASI during the two week AMoN measurement period and only IASI data within 25 km of the AMoN site are used. They then calculate trends from both datasets and show that they are comparable. Having established that IASI and AMoN data provide similar trends, they use the IASI data to calculate trends across the CONUS region, both annually and seasonally. They find that $NH_3$ is increasing faster than $10\%\cdot yr^{-1}$ in the eastern U.S. and Midwest in the spring and in the western U.S. in the summer. Trends in $NH_3$ "hotspots" (e.g., central Iowa), and urban areas (e.g. New York), which are mostly far from any AMoN site, are also shown to be positive and significant. The paper is very well organized and the results are certainly important. There are some sections that could be more clearly written, as I have detailed below. I recommend the paper be published with minor revisions.*

**Presentation issues**

Section 2.2: *the authors state that they used AMoN data for all sites except UT01. This is true for the correlation analysis, but not the trend analysis, which used a much more limited set, as is explained later. This should be made clear in section 2.3.2.*

Response: we added a sentence in section 2.2 to clarify the dataset used for trend analyses:

Line 215: We include AMoN trend analysis only for sites with full year coverage during 2008 - 2018 (N=13).

Section 2.3.1: *were the maps really created using the data for entire 2008-2018 period? Based on the discussion in section 4 it seems that they were created for each month of each year, which would be the only way to generate Figure 5. Or maybe multiple versions of the maps were created and used for different estimates? Please make this clear.*

Response: In Figure 5 (a), monthly averaged AMoN and IASI $NH_3$ time series within 25 km of the AMoN location are used for visualization. For the seasonal trend analyses, to achieve a higher spatial resolution (~ 2 km), seasonally oversampling maps are generated for spring (March, April, and May, MAM),

summer (June, July, and August, JJA), fall (September, October, and November, SON), and winter (December, January, and February, DJF) for each year. We've changed the text in section 4.1 to clarify the different versions of maps used in the manuscript.

Fig. 5 shows monthly averaged IASI and AMoN time series from Indianapolis, Indiana, USA (IN 99).

Line 435 - 439: Here we will compare IASI $NH_3$ trends with the AMoN observed $NH_3$ trends in the CONUS over the last decade. We include AMoN trend analysis only for sites with full year coverage during 2008 - 2018 (N=13). To achieve a higher spatial resolution, in the following study, we used the oversampled IASI $NH_3$ maps for trend analyses at 2 km scale. A long-term trend analysis was then performed using AMoN and IASI oversampled data (Sun et al., 2018; Wang et al., 2021) by Theil-Sen's slope estimator and MK test to examine the agreement between the datasets and explore any regional differences.

Section 2.3.2: *the authors state that they use the Mann-Kendall test and the Theil-Sen slope estimator for trend analyses. This is a good approach, but never again do they mention either technique. The reader is left to assume that the Theil-Sen slope estimator is used in the trend analysis section, and it's not clear where the Mann-Kendall test is used at all, since all the correlations are presented as Pearson coefficients. If the Mann-Kendall test and the Theil-Sen slope estimator are only used in the trend section, please make this clear, and then provide a few sentences demonstrating how they are applied.*

Response: The Pearson's r values are used for the comparison between IASI and AMoN in Section 3. The MK test and Theil-Sen slope estimator are used only for trend analysis in Section 4. MK test is used to determine the significance level of observed trends. We've changed the text in Section 3. to clarify that.

Line 247 - 248: In this study, Theil-Sen's slope was used to estimate 2008 – 2018 $NH_3$ trends, and the MK test was used to derive the significance level of trends.

Line 437 - 439: A long-term trend analysis was then performed using AMoN and IASI oversampled data (Sun et al., 2018; Wang et al., 2021) by Theil-Sen's slope estimator and MK test to examine the agreement between the datasets and explore any regional differences.

Tables 2 and 3: *it seems that the first row captions have been switched between Tables 2 and 3. And please define pair. Does it mean all the IASI pixels co-located with an AMoN site during one two week period? It would be better to say: AMoN-IASI pair. Or add the following on line 238: ... for comparison, establishing an AMoN-IASI pair.*

Response: Table 2 and 3 show the impact of temporal coverage and number of IASI pixels, respectively. "Pair" means each AMoN sample with co-located IASI pixels. We've added the definition for pair and changed the text to AMoN-IASI pairs in Table 1, 2, 3, and 4 for clarification.

Line 255: We define each AMoN sample with co-located IASI pixels as an AMoN-IASI pair.

Section 3.3: *Please provide the correlations between IASI and AMoN for all four seasons and not just winter. Please clarify how Figure 4 was obtained. The text suggest that IASI data were oversampled over the entire 2008-2018 period. I don't see how correlations can be calculated, since such oversampled would have no temporal information. Was the oversampling done on a month by month basis? This point is key for understanding how the results in section 4 were obtained. I suggest a scatter plot of the AMoN median values and the Pearson coefficient. This would reinforce the conclusions in the paragraph starting at line 360.*

*The section in the last paragraph in section 3.3 starting at line 377 (However, …) jumps ahead to discuss results in the next section. I think the authors should move this text to the next section, and use the plot I suggested above here to make the points stated at the beginning of this paragraph.*

Response: We've added the correlations between IASI and AMoN for all four seasons. Figure 4 shows multi-year averaged $NH_3$ seasonality comparison results between AMoN sites and the IASI observations using 25 km spatial window, which means we ignore the interannual variability of $NH_3$ seasonality and calculate averaged IASI and AMoN $NH_3$ seasonality during 2008 – 2018. The oversampling products are only used for trend analyses in Section 4 to achieve a higher spatial resolution, not for the comparison results in Section 3. We've added the scatter plot suggested by the reviewer as Figure 4(b) and changed the text. We agree with the reviewer that having the seasonality regression plots in Section 4 causes confusion and we've rearranged the regression plots for individual AMoN sites to Figure 4.

Line 371 - 373: When temporal coverage is at least 80%, IASI wintertime data still have good agreement with AMoN (r = 0.61) although the comparison is limited to only a few AMoN & IASI pairs (N = 33). The r values for spring, summer, and autumn when temporal coverage $\geqslant$ 80% are 0.60 (N = 181), 0.76 (N = 502), and 0.70 (N = 283), respectively.

Line 229 - 231: The oversampling products are only used for the trend analyses in Section 4 to achieve a high spatial resolution. For IASI and AMoN comparison results in Section 3, the oversampling products are not used since it sacrifices the temporal resolution.

Line 435 - 438: Here we will compare IASI $NH_3$ trends with the AMoN observed $NH_3$ trends in the CONUS over the last decade. We include AMoN trend analysis only for sites with full year coverage during 2008 - 2018 (N = 13). To achieve a higher spatial resolution, in the following study, we used the oversampled IASI $NH_3$ maps to calculate $NH_3$ trend for each 2 km grid box.

[Figure]

**Figure 4. (a)** Multi-year averaged NH$_3$ seasonality comparison results between AMoN sites and the IASI observations within 25 km of the AMoN sites at monthly resolution. Circles without filled color denote the AMoN sites with no statistically significant correlation with IASI ($\alpha = 0.05$). The circle sizes denote the length of AMoN data record; **(b)** The relationship between mean AMoN NH$_3$ concentrations and the correlation between AMoN and IASI seasonality; The regression between IASI and AMoN observed NH$_3$ seasonality for **(c)** the AMoN site in Joshua Tree National Park, California (CA67), **(d)** the AMoN site in Sequoia National Park, California, and **(e)** the AMoN site in Indianapolis, Indiana (IN99).

Section 4: *Please provide more detail on the calculation of the trends: were the trends calculated on each 2 km grid box, how long were the averaging time periods, how were the IASI trends averaged to compare with AMoN trends, how were the IASI data averaged to provide regional, hotspot and CONUS trends. This information is critical for understanding the results discussed in section 4.1 and 4.2. For example, it's hard to understand how the CONUS trend can be 3.9%/yr (line 428), 8.0%/yr (line 436) and 6.8%/yr (line 528). Either there is an error or these three values are calculated differently, but it's not clear what the difference is.*

Response: The trends were calculated on 2 km oversampling grid box. For the annual trends, oversampled maps were generated for each year and for the seasonal trends, oversampled maps were generated for each year for each season. For the comparison between IASI and AMoN trends, a 25 km spatial window is used. We used two different approaches to show the importance of high resolution NH$_3$ maps. The first method is considering the CONUS as a whole to calculate an averaged NH$_3$ concentration for each year to perform trend analysis, which is 3.9%· yr$^{-1}$. The second method is calculating NH$_3$ trends for each 2 km

grid box and then use the median value of all grid boxes to represent CONUS trend, which is 6.8%· yr$^{-1}$. The value 8.0%· yr$^{-1}$ was a typo and should be 6.8%· yr$^{-1}$, and we apologize for the typo that caused the confusion. We've changed the text for clarification and corrected the numbers for trend analyses.

Line 221 - 231: From 2008 to 2018, a 0.02° × 0.02° (~2 km) annual mean NH$_3$ map in the CONUS was created each year based on a physical oversampling algorithm that represents the satellite spatial response functions as generalized 2-D super Gaussian functions (Sun et al., 2018). This algorithm weighs IASI measurements by their uncertainties, which include varying sensitivities to thermal contrast as described in Sun et al. (2018) and Wang et al. (2021). To evaluate the seasonal trends, for each year, seasonally averaged oversampling maps were also generated for spring (March, April, and May, MAM), summer (June, July, and August, JJA), fall (September, October, and November, SON), and winter (December, January, and February, DJF). For each season, we were able to achieve sufficiently overlapped IASI pixels through calculating the sum of the unnormalized spatial response function (SRF) of the oversampling results. A large sum of unnormalized SRF means the Level 3 grid is covered by more Level 2 pixels. Sun et al. 2018 and Wang et al. 2021 have a detailed description of SRF. The oversampling products are only used for the trend analyses in Section 4 to achieve a high spatial resolution. For IASI and AMoN comparison results in Section 3, the oversampling products are not used since it scarifies the temporal resolution.

Line 446 - 447: When plotting the trends of AMoN sites against the median of IASI trends within a 25 km spatial window (Fig. 7), a moderate correlation (r = 0.66) was found between IASI and AMoN NH$_3$ trends.

Line 459 - 463: Here we note that the spatial resolution could affect the results of trend analyses. The trend 6.8% · yr$^{-1}$ was derived as the median of trends for each 2 km grid box. If considering the CONUS as a whole and calculating the annual mean NH$_3$ for the whole CONUS during 2008 – 2018 to derive the overall trend in CONUS, the IASI NH$_3$ change for 2008 – 2018 is (3.9 ± 2.2) % · yr$^{-1}$ and (1.3 ± 0.8) × 10$^{14}$ molec/cm$^2$·yr$^{-1}$, similar with the trend in the previous study (3.4 ± 0.6) % · yr$^{-1}$ and (1.1 ± 0.4) × 10$^{14}$ molec/cm$^2$·yr$^{-1}$) (Van Damme et al., 2021).

Line 472: Although the percent changes in the regions with the highest concentrations are smaller than the trend in the CONUS median (6.8% · yr$^{-1}$) …

Line 537: The urban areas have a similar NH$_3$ trend compared with CONUS (8.1% · yr$^{-1}$ vs. 6.8% · yr$^{-1}$) …

Figure 5: *It would be interesting to show this figure for a few more sites; this would provide the reader with an idea of the variability in the slope and correlation; the sites mentioned at the end of section 3 (CA67 and CA 83) should be shown and could be discussed here.*

Response: We've added the regression plots for CA67 and CA83 as Figure 4(c) and (d).

[Figure]

**Figure 4. (a)** Multi-year averaged NH$_3$ seasonality comparison results between AMoN sites and the IASI observations within 25 km of the AMoN sites at monthly resolution. Circles without filled color denote the AMoN sites with no statistically significant correlation with IASI ($\alpha$ = 0.05). The circle sizes denote the length of AMoN data record; **(b)** The relationship between mean AMoN NH$_3$ concentrations and the correlation between AMoN and IASI seasonality; The regression between IASI and AMoN observed NH$_3$ seasonality for **(c)** the AMoN site in Joshua Tree National Park, California (CA67), **(d)** the AMoN site in Sequoia National Park, California, and **(e)** the AMoN site in Indianapolis, Indiana (IN99).

Figure 6: *are the IASI trends calculated in each 2 km grid box? What is the spatial extent of each "hotspot" box?*
Response: IASI trends are calculated in each 2 km grid box. We use Hoshen–Kopelman algorithm to cluster adjacent grid points above the 95$^{th}$ percentile threshold as a hotspot (Hoshen & Kopelman, 1976), and the median area of identified hotspots is ~ 150 km$^2$ (Wang et al., 2021).

Line 436 - 437: To achieve a higher spatial resolution, in the following study, we used the oversampled IASI NH$_3$ maps to calculate NH$_3$ trend for each 2 km grid box.
Line 466 - 469: We use Hoshen–Kopelman algorithm to cluster adjacent grid points above the 95$^{th}$ percentile threshold of the 11-year CONUS oversampling map (6.7 × 10$^{15}$ molec/cm$^2$) as a NH$_3$ hotspot (Hoshen & Kopelman, 1976; Wang et al., 2021), and the median area of identified hotspots is ~ 150 km$^2$ (Wang et al., 2021).

**Minor revisions**

Line 17: *The limited number of NH₃ observations hinders …*
Response: We've changed the text based on the reviewer's suggestion:
Line 17: The limited number of NH₃ observations hinders…

Line 19: *…networks are few and sparse across most of the globe,*
Response: We've changed the text based on the reviewer's suggestion:
Line 18: are few and sparse across most of the globe…

Line 34: *(2008-2018), suggesting the NH₃ will become a greater contributor to nitrogen deposition. NH₃ trends at AMoN sites are correlated with IASI NH₃ trends (r=0.6), and show similar spatial patterns, with the highest increases in the Midwest and eastern U.S*
Response: We've changed the text based on the reviewer's suggestion:
Line 33 - 35: suggesting the NH₃ will become a greater contributor to nitrogen deposition. NH₃ trends at AMoN sites are correlated with IASI NH₃ trends (r = 0.66), and show similar spatial patterns, with the highest increases in the Midwest and eastern U.S.

Line 37: *respectively. NH₃ hotpots (defined as regions where the IASI NH₃ column is larger than the 95ᵗʰ percentile of 11-year CONUS map, 6.7 Å~ 10¹⁵ molec/cm², are also experiencing increasing concentrations over time, with a median of NH₃ trend of 4.7% · yr⁻¹.*
Response: We've changed the text based on the reviewer's suggestion:
Line 39 - 42: NH₃ hotpots (defined as regions where the IASI NH₃ column is larger than the 95ᵗʰ percentile of 11-year CONUS map, $6.7 \times 10^{15}$ molec/cm²), also experiencing increasing concentrations over time, with a median of NH₃ trend of 4.7% · yr⁻¹.

Line 42: *The increases in NH₃ …*
Response: We've changed the text based on the reviewer's suggestion:
Line 43: The increases in NH₃

Line 43: *areas, and therefore should be carefully monitored and studied.*
Response: We've changed the text based on the reviewer's suggestion:
Line 44 - 45: areas, and therefore should be carefully monitored and studied.

Line 52: *NH₄NO₃*
Response: We've changed the text based on the reviewer's suggestion:
Line 53: NH₄NO₃

Line 55: *emissions are decreasing under new pollution controls*
Response: We've changed the text based on the reviewer's suggestion:
Line 60: NOₓ emissions are decreasing under new pollution controls

Line 58: *in sensitive ecosystems.*
Response: We've changed the text based on the reviewer's suggestion:
Line 62: especially in sensitive ecosystems

Line 79: *Move definition of IMPROVE to this line.*
Response: We've moved the definition of IMRPOVE:
Line 83: synthesizing the AMoN NH$_3$ data with other ground monitoring networks, e.g., the Interagency Monitoring of Protected Visual Environments (IMPROVE)

Line 85: *hotspot regions*
Response: We've changed the text based on the reviewer's suggestion:
Line 90: hotspot regions

Line 94: *(S-NPP) and on JPSS-1 and JPSS-2*
Response: We've changed the text based on the reviewer's suggestion:
Line 99 – 100: Cross-track Infrared Sounder (CrIS) on NOAA and NASA Suomi National Polar-orbiting Partnership (S-NPP) and on Joint Polar Satellite System-1 and -2 (JPSS-1 and -2)

Line 113: *validation efforts were carried out in specific seasons*
Response: We've changed the text based on the reviewer's suggestion:
Line 118: All of these validation works were carried out in specific seasons

Line 126: *in the atmosphere is limited*
Response: We've changed the text based on the reviewer's suggestion:
Line 133: in the atmosphere is limited

Line 146: *in more detail.*
Response: We've changed the text based on the reviewer's suggestion:
Line 152: needs to be examined in more detail.

Line 167: *on bi-weekly/seasonal*
Response: We've changed the text based on the reviewer's suggestion:
Line 174: between IASI and AMoN on bi-weekly/seasonal scales

Line 169: *We avoided converting column NH$_3$ into surface concentrations*
Response: We've changed the text based on the reviewer's suggestion:
Line 175 - 176: We avoided converting column NH$_3$ into surface concentrations…

**Line 178:** *sounder deployed on board*
Response: We've changed the text based on the reviewer's suggestion:
Line 184: IASI is an infrared sounder deployed on board of

**Line 183:** *The latest version is a reanalysis dataset that uses the European ....*
Response: We've changed the text based on the reviewer's suggestion:
Line 189 - 190: The latest version is a reanalyzed dataset that uses the

**Line 185:** *Because these meteorological data are coherent in time, the reanalyzed $NH_3$ dataset is the most appropriate one to study trends.*
Response: We've changed the text based on the reviewer's suggestion:
Line 191- 192: Because these meteorological data are coherent in time, the reanalysis dataset it is the most appropriate dataset to study trends.

**Line 200:** *imagery, we determined*
Response: We've changed the text based on the reviewer's suggestion:
Line 207: Using satellite imagery, we determined that

**Line 214:** *this algorithm weights IASI measurements by their uncertainties, which include varying sensitivities to thermal contrast, as described*
Response: We've changed the text based on the reviewer's suggestion:
Line 223 - 224: This algorithm weights IASI measurements by their uncertainties, which include varying sensitivities to thermal contrast as described in

**Line 218:** *The sentence: "For each season, we were able to achieve sufficiently overlapped IASI pixels through calculating the sum of the unnormalized spatial response function (SRF) of the oversampling results" is not clear. Please clarify: a bit more detail might be helpful.*
Response: We've modified the sentence to clarify the definition of SRF:
Line 226 - 230: For each season, we were able to achieve sufficiently overlapped Level 2 IASI pixels for each Level 3 grid through calculating the sum of the unnormalized spatial response function (SRF) of the oversampling results. A large sum of unnormalized SRF means the Level 3 grid is covered by more Level 2 pixels. Sun et al. 2018 and Wang et al. 2021 have a detailed description of SRF.

**Line 229:** *Unlike simple linear regressions,*
Response: We've changed the text based on the reviewer's suggestion:
Line 241: Unlike simple linear regression

**Line 236:** *For the initial analysis, we first used the simplest method for comparing ...*
Response: We've changed the text based on the reviewer's suggestion:

Line 252: For the initial analysis, we first used the simplest method for comparing the satellite measurements with ground observations.

Line 237: *for each AMoN site, we average all IASI observations within a given radius of the site during the AMoN sampling time frame (2 weeks)*
Response: We've changed the text based on the reviewer's suggestion:
Line 253 - 254: In other words, for each AMoN site, we average all IASI observations within a given radius of the AMoN site during the sampling time frame (2 weeks) for comparison and refer to that radius as a spatial window.

Line 239: *between the two datasets*
Response: We've changed the text based on the reviewer's suggestion:
Line 256: between the two datasets

Line 250: *and the number of IASI pixels*
Response: We've changed the text based on the reviewer's suggestion:
Line 267: and the number of IASI pixels

Line 254: *or in regions*
Response: We've changed the text based on the reviewer's suggestion:
Line 267: or in regions

Figure 1: *label each plot with the AMoN site location*
Response: All of these example plots are based on data from one AMoN site. We've added the AMoN site location in the figure caption:
Line 297: Figure 1. Examples of IASI data temporal coverage over the biweekly AMoN sampling period for an AMoN site in Yosemite National Park, California (CA 44).

Line 267: *the 2-week AMoN integration period, (using a 25 km spatial window), could affect the results.*
Response: We've changed the text based on the reviewer's suggestion:
Line 284: the 2-week AMoN integration period (using a 25 km spatial window), could affect the results.

Line 276: *as representative as*
Response: We've changed the text based on the reviewer's suggestion:
Line 292: to be as representative as

Line 281: *for most days*
Response: We've changed the text based on the reviewer's suggestion:

Line 298: for most days of

Line 285: *To this end, we explore the dependence of the correlation between IASI and AMoN on the IASI data temporal coverage of the 2-week sampling period and total number of IASI pixels within the 2-week AMoN sampling period, using the 25 km spatial window.*
Response: We've changed the text based on the reviewer's suggestion:
Line 301 – 302: To this end, we explore the dependence of the correlation between IASI and AMoN on IASI data's temporal coverage of the 2-week sampling period and total number of IASI pixels within the 2-week AMoN sampling period, using the 25 km spatial window.

Line 288: *The impact of temporal coverage and the number of IASI pixels within the sampling period are*
Response: We've changed the text based on the reviewer's suggestion:
Line 304: The impact of different temporal averaging and the number of IASI pixels

Line 314: *For each AMoN site, we repeated the two different sampling strategies 100 times*
Response: We've changed the text based on the reviewer's suggestion:
Line 332: For each AMoN site, we repeated the two different sampling strategies 100 times

Line 361: *Temporal averaging and regridding approaches, such as tessellation oversampling and physical oversampling, are common methods used to achieve higher spatial resolution*
Response: We've changed the text based on the reviewer's suggestion:
Line 381 - 383: The temporal averaging and regridding approaches, such as the tessellation oversampling and physical oversampling, are common methods to achieve higher spatial resolution by sacrificing the temporal resolution.

Line 364: *neglect the interannual variability and calculate the multi-year averaged IASI NH3 concentrations, both annual and seasonal, using the 25 km*
Response: Figure 4 shows multi-year averaged $NH_3$ seasonality comparison results between AMoN sites and the IASI observations using 25 km spatial window, which means we ignore the interannual variability of $NH_3$ seasonality and calculate averaged IASI and AMoN $NH_3$ seasonality during 2008 – 2018. We've changed the text based on the reviewer's suggestion:
Line 384 - 386: Here we neglect the interannual variability of $NH_3$ seasonality and calculate averaged IASI and AMoN $NH_3$ seasonality during 2008 - 2018 using the 25 km spatial window.

Line 366: *coverage and numbers of IASI pixels increase*
Response: We've changed the text based on the reviewer's suggestion:
Line 386: both temporal coverage and numbers of IASI pixels increase.

Line 393: *The methodology and comparison results in section 3 demonstrate that IASI $NH_3$ can*

*be used to verify and augment regional NH₃ trends over the last decade. Here we will compare IASI NH₃ trends with the AMoN observed NH₃ trends in the CONUS region over the last decade.*

Response: We've changed the text based on the reviewer's suggestion:

Line 420 - 421: The methodology and comparison results in section 3 demonstrate that IASI NH$_3$ can be used to verify and augment regional NH$_3$ trends over the last decade.

Line 398: *from Indianopolis*

Response: We've changed the text based on the reviewer's suggestion:

Line 426: from Indianapolis

Figure 5: *shouldn't the caption read "2008-2018 trends in monthly averaged NH₃"?*

Response: Yes. We have changed the as appropriate:

Line 432: 2008 – 2018 trends in monthly averaged NH$_3$

Line 417: *Remove the sentence starting with "The absolute" as this information is presented in the next paragraph.*

Response: We've deleted the text based on the reviewer's suggestion.

Line 433: *Analyzing NH₃ hotspots,*

Response: We've changed the text based on the reviewer's suggestion:

Line 468: Analyzing NH$_3$ hotpots

Line 434: *indicating that the regions with the largest emissions are also seeing concentrations increasing with time.*

Response: We've changed the text based on the reviewer's suggestion:

Line 470: indicating that the regions of the largest emissions sources are also seeing increasing concentrations over time.

Line 436: *smaller that the trend in the CONUS median*

Response: We've changed the text based on the reviewer's suggestion:

Line 472: are smaller than the trend in the CONUS median.

Line 437: *higher than the trend in the CONUS median*

Response: We've changed the text based on the reviewer's suggestion:

Line 473: are higher than the trend in the CONUS median.

Line 438: *define column-areal weighting*

Response: We've changed the text based on the reviewer's suggestion:

Line 474: The top 10 NH$_3$ hotspots in CONUS regarding column-areal weighting (NH$_3$ column times the area)

Line 441: *see the smallest changes.*
Response: We've changed the text based on the reviewer's suggestion:
Line 478: see the smallest changes.

Line 456: *in the eastern US*
Response: We've changed the text based on the reviewer's suggestion:
Line 492: in the eastern US

Line 457: *western US and the Northeast*
Response: We've changed the text based on the reviewer's suggestion:
Line 493: the Northeast US

Line 487: *add some text like: (Wang et al., 2021), a gap that IASI data can fill.*
Response: We've changed the text based on the reviewer's suggestion:
Line 525: are not located in highly populated urban regions (Wang et al., 2021), a gap that IASI data can fill.

Line 488: *the cumulative distribution of the CONUS population as a function of the distance from an AMoN site.*
Response: We've changed the text based on the reviewer's suggestion:
Line 526: Fig. 9a shows the cumulative distribution of the US population as a function of the distance from an AMoN site.

Line 493: *mobile sources and trends in population centers.*
Response: We've changed the text based on the reviewer's suggestion:
Line 531: from mobile sources and trends in population centers.

Line 502: *in total account for more than seventy million people*
Response: We've changed the text based on the reviewer's suggestion:
Line 540: in total account for more than seventy million people

Figure 9: *Cumulative distribution of CONUS population as a function of distance from an AMoN site.*
Response: We've changed the text based on the reviewer's suggestion:
Line 558: Figure 9. (a) Cumulative distribution of CONUS population as a function of distance from the nearest AMoN site

Line 517: *The temporal coverage of the IASI data during the two week AMoN sampling period is the controlling factor of the correlation between the IASI and AMoN measurements, presumably because of the large day-to-day variability of NH₃.*

Response: We've changed the text based on the reviewer's suggestion:

Line 570 - 572: The temporal coverage of IASI data during the 2-week AMoN sampling period is the controlling factor of the correlation between IASI and AMoN measurements, presumably because of the large day-to-day variability of $NH_3$.

Line 521: *the IASI NH₃ product*

Response: We've changed the text based on the reviewer's suggestion:

Line 574: the IASI $NH_3$ product

Line 522: *shown the unique role*

Response: We've changed the text based on the reviewer's suggestion:

Line 577: has shown the unique role

Line 530: *as well as similar spatial patterns.*

Response: We've changed the text based on the reviewer's suggestion:

Line 584: as well as similar spatial patterns

Line 531: show the largest increases in the Midwest and eastern U.S., with a moderate correlation between the IASI and AMoN trends for the entire CONUS.

Response: We've changed the text based on the reviewer's suggestion:

Line 585 – 586: Both IASI and AMoN show largest $NH_3$ increases in the Midwest and eastern U.S., with a moderate correlation between the IASI and AMoN trends for the entire CONUS (r = 0.66).

Line 535: *deposition in most regions in the U.S.;*

Response: We've changed the text based on the reviewer's suggestion:

Line 589: deposition in most regions in the U.S.

Line 536: *(2016), which will have adverse impacts*

Response: We've changed the text based on the reviewer's suggestion:

Line 591: which will have adverse impacts

Line 540: *(cropland dominated) and in summer in the western U.S. (feedlot dominated), highlighting the impacts*

Response: We've changed the text based on the reviewer's suggestion:

Line 595: in the Midwest and eastern U.S. (cropland dominated) and in summer in the western U.S. (feedlot dominated)

Line 543: *for characterizing NH₃ magnitude*
Response: We've changed the text based on the reviewer's suggestion:
Line 599: for characterizing NH$_3$ magnitude

Line 547: *communities with limited resources*
Response: We've changed the text based on the reviewer's suggestion:
Line 602: communities with limited resources

---

## Author Comment (AC2)

Response to Review#2
Title: "Bridging the spatial gaps of the Ammonia Monitoring Network using satellite ammonia measurements"
Authors: R. Wang et al.

We thank the reviewer for the helpful comments. We have revised the manuscript accordingly to help clarify and focus the manuscript. The original comments from reviewers are in *blue and italics*, our replies are in black font, and verbatim responses from the revised manuscript are in red font.

**Summary**

*This study is well-organized and important for research community. This reviewer has two minor comments. 1) ? 2) can urban atmospheric NH3 prevent the previously formed NH4NO3 from evaporating? The current discussion is imbalance.*

**Minor comments:**

Minor comment #1: *the authors should explain why only the data ended on 2018, but not 2022*

Response: IASI observations in 2020 were excluded to rule out the possible impact of the pandemic. IASI observations in 2021 and 2022 were not included because Metop-A retired in 2021.

Minor comment #2: *can urban atmospheric $NH_3$ prevent the previously formed $NH_4NO_3$ from evaporating? The current discussion is imbalance.*

Response: The equilibrium between gas phase $NH_3$ and $HNO_3$ and aerosol phase $NH_4NO_3$ shifts to the aerosol phase at cold temperature and high particle pH condition (Feng et al., 2020; Guo et al., 2018; Shah et al., 2018). Gas phase $NH_3$ also plays an important role in aerosol acidity and aerosol chemistry (Lawal et al., 2018). Zhai et al. 2021 demonstrates that gas phase $NH_3$ hinders the scavenging of nitrate aerosol by slowing down the deposition of total inorganic nitrate. Wang et al. 2020 shows that gas phase $NH_3$ and $HNO_3$ can nucleate directly to form $NH_4NO_3$ particles in cold atmospheric conditions and is likely to result in rapid growth of new atmospheric particles in winter, especially in urban environments with abundant $HNO_3$. Unfortunately, reliable gas phase $HNO_3$ data in the boundary layer are not readily available to make a full evaluation of $NH_4NO_3$ formation. Because trends in $SO_2$ and $NO_x$ also impact $NH_4NO_3$ formation, it is difficult to evaluate how each city may respond to increases in $NH_3$ over time. We've added Fig. 9b and the following paragraph to discuss the $NH_3$'s role in $NH_4NO_3$ formation in urban areas in details:

[Figure]

**Figure 9.** **(a)** Cumulative distribution of CONUS population as a function of distance from the nearest AMoN site; **(b)** Correlation between EPA NH$_3$ emissions and IASI observed mean NH$_3$ concentrations at state level during 2008 - 2018. The gray dots represent states without statistically significant correlations (α = 0.05).

Line 541 - 554: The urban environment with abundant HNO$_3$ and NH$_3$ emissions from vehicles nominally favors the formation of NH$_4$NO$_3$ over rural areas. Recent studies suggest that gas phase NH$_3$ hinders the scavenging of NH$_4$NO$_3$ by slowing down the deposition process of total inorganic nitrate (Zhai et al., 2021) and promotes new atmospheric particle formation by directly nucleate with HNO$_3$ to form NH$_4$NO$_3$ in winter in urban areas and (Wang et al., 2020). However, ultimately the sensitivity to PM$_{2.5}$ from increases in NH$_3$ in any urban areas will be a complex function of trends of NO$_x$ and SO$_2$ as well (Feng et al., 2020). The NH$_3$ increase in these densely populated areas and its impact on  aerosol chemistry needs to be further addressed. For example, Fig. 9b shows the relationship between NH$_3$ trends versus emissions trends (EPA Air Pollutant Emissions Trends Data) on the state level. For agricultural areas with high NH$_3$ (excess NH$_3$ relative to NH$_4$NO$_3$ equilibrium), one would expect an increase in emissions to correlate very well with increasing NH$_3$ columns. In contrast, in areas with more NO$_x$, increases in emissions may result in NH$_3$ going into NH$_4$NO$_3$ and thereby show little or even negative correlations. To this end, Fig. 9b shows that at state level, agricultural states show strong correlations between emissions and concentrations trends, e.g., California, while the more urbanized northeast states show weak or negative correlations, e.g., New Jersey. Ultimately, co-located aerosol phase and gas phase precursor measurements are needed to fully deduce what is happening at each urban area and should be a focus of future air quality network integration.

Line 607 - 610: The comparison between NH$_3$ emission trends and IASI observed NH$_3$ concentration trends suggests that strong correlations exist in agricultural states, e.g., California, while weak or negative correlations in more urbanized northeast states, e.g., New Jersey, indicating the different contribution from emission and partitioning.

Line 43 - 45: A comparison between IASI NH$_3$ concentration trends and state-level NH$_3$ emission trends is then performed to reveal that good correlations exist in agricultural states while negative correlations in more urbanized states, suggesting the different roles of emission and partitioning in NH$_3$ increases.

**References:**

EPA, United States Environmental Protection Agency, Air Pollutant Emissions Trends Data, https://www.epa.gov/air-emissions-inventories/air-pollutant-emissions-trends-data, last access: August 2023.

Feng, J., Chan, E., and Vet, R.: Air quality in the eastern United States and Eastern Canada for 1990–2015: 25 years of change in response to emission reductions of SO2 and NOx in the region, Atmos. Chem. Phys., 20, 3107–3134, https://doi.org/10.5194/ACP-20-3107-2020, 2020.

Guo, H., Otjes, R., Schlag, P., Kiendler-Scharr, A., Nenes, A., and Weber, R. J.: Effectiveness of ammonia reduction on control of fine particle nitrate, Atmos. Chem. Phys., 18, 12241–12256, https://doi.org/10.5194/acp-18-12241-2018, 2018.

Lawal, A. S., Guan, X., Liu, C., Henneman, L. R. F., Vasilakos, P., Bhogineni, V., Weber, R. J., Nenes, A., and Russell, A. G.: Linked Response of Aerosol Acidity and Ammonia to $SO_2$ and $NO_x$ Emissions Reductions in the United States, Environ. Sci. Technol., https://doi.org/10.1021/acs.est.8b00711, 2018.

Shah, V., Jaeglé, L., Thornton, J. A., Lopez-Hilfiker, F. D., Lee, B. H., Schroder, J. C., Campuzano-Jost, P., Jimenez, J. L., Guo, H., Sullivan, A. P., Weber, R. J., Green, J. R., Fiddler, M. N., Bililign, S., Campos, T. L., Stell, M., Weinheimer, A. J., Montzka, D. D., and Brown, S. S.: Chemical feedbacks weaken the wintertime response of particulate sulfate and nitrate to emissions reductions over the eastern United States, Proc. Natl. Acad. Sci. U.S.A., https://doi.org/10.1073/pnas.1803295115, 2018.

Wang, M., Kong, W., Marten, R., He, X. C., Chen, D., Pfeifer, J., Heitto, A., Kontkanen, J., Dada, L., Kürten, A., Yli-Juuti, T., Manninen, H. E., Amanatidis, S., Amorim, A., Baalbaki, R., Baccarini, A., Bell, D. M., Bertozzi, B., Bräkling, S., Brilke, S., Murillo, L. C., Chiu, R., Chu, B., de Menezes, L. P., Duplissy, J., Finkenzeller, H., Carracedo, L. G., Granzin, M., Guida, R., Hansel, A., Hofbauer, V., Krechmer, J., Lehtipalo, K., Lamkaddam, H., Lampimäki, M., Lee, C. P., Makhmutov, V., Marie, G., Mathot, S., Mauldin, R. L., Mentler, B., Müller, T., Onnela, A., Partoll, E., Petäjä, T., Philippov, M., Pospisilova, V., Ranjithkumar, A., Rissanen, M., Rörup, B., Scholz, W., Shen, J., Simon, M., Sipilä, M., Steiner, G., Stolzenburg, D., Tham, Y. J., Tomé, A., Wagner, A. C., Wang, D. S., Wang, Y., Weber, S. K., Winkler, P. M., Wlasits, P. J., Wu, Y., Xiao, M., Ye, Q., Zauner-Wieczorek, M., Zhou, X., Volkamer, R., Riipinen, I., Dommen, J., Curtius, J., Baltensperger, U., Kulmala, M., Worsnop, D. R., Kirkby, J., Seinfeld, J. H., El-Haddad, I., Flagan, R. C., and Donahue, N. M.: Rapid growth of new atmospheric particles by nitric acid and ammonia condensation, Nature, 581, 184–189, https://doi.org/10.1038/s41586-020-2270-4, 2020.

Zhai, S., Jacob, D. J., Wang, X., Liu, Z., Wen, T., Shah, V., Li, K., Moch, J. M., Bates, K. H., Song, S., Shen, L., Zhang, Y., Luo, G., Yu, F., Sun, Y., Wang, L., Qi, M., Tao, J., Gui, K., Xu, H., Zhang, Q., Zhao, T., Wang, Y., Lee, H. C., Choi, H., and Liao, H.: Control of particulate nitrate air pollution in China, Nat. Geosci., 14, 389–395, https://doi.org/10.1038/s41561-021-00726-z, 2021.